# A STIM dependent dopamine-neuropeptide axis maintains the larval drive to feed and grow in Drosophila

Nandashree Kasturacharya[1,2], Jasmine Kaur Dhall[1], Gaiti Hasan [1,2]*

**1** National Centre for Biological Sciences, TIFR, Bellary Road, Bengaluru, India, **2** The University of Trans-Disciplinary Health Sciences and Technology (TDU), Bengaluru, India

* gaiti@ncbs.res.in

**Data Availability Statement:** All relevant data are within the paper and its Supporting Information files except the complete data for RNAseq which is available at https://www.ncbi.nlm.nih.gov/geo/query/acc.cgi?acc=GSE202098.

## Abstract

Appropriate nutritional intake is essential for organismal survival. In holometabolous insects such as *Drosophila melanogaster*, the quality and quantity of food ingested as larvae determines adult size and fecundity. Here we have identified a subset of dopaminergic neurons (THD') that maintain the larval motivation to feed. Dopamine release from these neurons requires the ER Ca$^{2+}$ sensor STIM. Larvae with loss of STIM stop feeding and growing, whereas expression of STIM in THD' neurons rescues feeding, growth and viability of STIM null mutants to a significant extent. Moreover STIM is essential for maintaining excitability and release of dopamine from THD' neurons. Optogenetic stimulation of THD' neurons activated neuropeptidergic cells, including median neuro secretory cells that secrete insulin-like peptides. Loss of STIM in THD' cells alters the developmental profile of specific insulin-like peptides including ilp3. Loss of ilp3 partially rescues STIM null mutants and inappropriate expression of ilp3 in larvae affects development and growth. In summary we have identified a novel STIM-dependent function of dopamine neurons that modulates developmental changes in larval feeding behaviour and growth.

## Author summary

The ability to feed appropriately when hungry is an essential feature for organismal survival and is under complex neuronal control. An array of neurotransmitters and neuropeptides integrate external and internal signalling cues to initiate, maintain and terminate feeding. In adult vertebrates and invertebrates dopamine serves as a reward cue for motor actions, including feeding. Larvae of holometabolous insects, including *Drosophila melanogaster*, feed and grow constantly followed by gradual cessation of feeding, once sufficient growth is achieved for transition to the next stages of development. Here we identified a subset of larval dopaminergic neurons in *Drosophila melanogaster*, activity in which maintains continuous feeding in larvae. By analysis of a null mutant we show that these neurons require the Stromal Interaction Molecule (STIM) an ER Ca$^{2+}$ sensor, to maintain excitability. In turn they modulate activity of certain neuropeptidergic cells. Among these are the median neurosecretory cells (MNSc) that synthesize and secrete

**Funding:** This study was supported by core funds from NCBS (TIFR), a SERB distinguished fellowship to GH (SB/S9/YSCP/SERB-DF/2018-1), a DBT grant (BT/PR28450/MED/122/166/2018) to GH and a DST-WOS-A grant (SR/WOS-A/LS-703/2016) to NK. The funders had no role in study design, data collection and analysis, decision to publish, or preparation of the manuscript.

**Competing interests:** The authors have declared that no competing interests exist.

insulin-like peptides including ilp3. Loss of activity in the identified dopaminergic neurons dysregulates the normal pattern of larval ilp3 expression that correlates with altered growth. Overall, our study identified a simple dopamine modulated mechanism for feeding and growth whose manipulation could be useful for model organism studies related to feeding disorders, obesity and diabetes.

## Introduction

Animal growth occurs primarily during the juvenile stage of development. In holometabolous insects, including *Drosophila*, larval development is considered equivalent to the juvenile stage [1,2]. Steady and appropriate nutritional intake in larvae, is essential for growth and development, and ultimately determines both survival and fecundity of the animals. In *Drosophila*, the feeding rate increases in the second instar larval stage and larvae continue to feed voraciously till the wandering stage of third instar larvae [3]. Increase in feeding rate is accompanied by acceleration of cell division and cell growth [4].

Feeding behaviour and its modulation in *Drosophila* larvae has been studied primarily in the third instar larval stage, where it is regulated by multiple neurotransmitters and neuropeptides that respond to both external cues and the internal metabolic state [5]. Neuropeptide F (NPF; a human NPY homolog) serves as a motivational signal for foraging in larvae in response to appetitive odours. The activity of NPF neurons appears dependent on inputs from two pairs of central dopaminergic neurons that receive tertiary olfactory inputs [6,7]. Mutants for the short neuropeptide F (sNPF), encoded by an independent gene from NPF, affect body size by regulating food intake in larvae [8]. When food is restricted octopaminergic circuits regulate feeding independent of NPF signalling in 3rd instar larvae [9]. Neurons that secrete the Hugin neuropeptide respond to averse gustatory signals and their activation suppresses larval feeding [10]. Where essential amino acids are imbalanced a subset of dopaminergic neurons are required for rejection of food by larvae [11]. In addition serotonergic neurons from the brain project to the gut where they potentially regulate feeding-related muscle movements [12]. Insulin like peptides (ilps), are secreted by the medial neurosecretory cells (MNSc) that access the internal metabolic state and release ilps into neurohaemal sites for circulation. In adults ilps terminate feeding based on the energy state of the organism [13]. Analysis of the recently concluded larval connectome demonstrates that MNSc receive both direct and indirect inputs from the enteric nervous system found in the larval gut [14]. The larval MNSc also receive synaptic inputs from central neurons that release the Hugin neuropeptide [10]. Ilps released through neurohaemal sites from the MNSc circulate through the body and regulate energy metabolism [15], synthesis and release of the steroid hormone ecdysone from the prothoracic gland which drives larval instar progression [16,17].

A key difference between larval and adult feeding behaviour is that adult *Drosophila* feed sporadically, driven by hunger and satiety signals [18] whereas *Drosophila* larvae accelerate feeding as second instar larvae and feed continuously till the wandering stage of third instar larvae to optimise growth. They stop feeding as wandering larvae for a few hours prior to pupariation [19,20]. Despite studies identifying several neurotransmitters and neuropeptides in feeding regulation and their cognate neurons as part of the feeding connectome in larvae [5] mechanisms that initiate and maintain persistent feeding in early larval stages are not fully understood. In this study, whilst characterizing the cellular and molecular phenotypes of null mutants for the ER-Calcium sensor protein STIM (**St**romal **I**nteraction **M**olecule) [21] we identified a novel dopaminergic-neuropeptide connection in the absence of which early larvae

feed poorly and grow slowly. Growth deficits and lethality in STIM mutants appeared to have a focus in dopaminergic cells [22]. However, the cellular basis of STIM function and systemic phenotypes arising from loss of STIM in dopaminergic neurons remained to be understood. Here, we show that STIM function is required in a subset of central dopaminergic neurons for their excitability and dopamine release. These dopaminergic neurons impact larval growth by providing the motivation for persistent larval feeding and modulating neuropeptidergic cells, including the MNSc, to regulate levels of insulin-like peptides.

## Results

### Reduced food intake and growth deficits in STIM mutant larvae

*STIM^{KO}* larvae appear normal after hatching but their transition from first to second instar stages is slower than wild-type animals (**S1A and S1B Fig**) [22] and as second instars they die gradually between 86h to 326h after egg laying (**AEL; S1B Fig**). To identify the precise time window when *STIM^{KO}* larvae become sickly they were observed over 6h time intervals from 36h to 90h AEL. Whereas, wild type (*Canton-S* or CS) larvae transition from 1^{st} to 2^{nd} instar between 42-54h AEL, the same transition in *STIM^{KO}* larvae occurs between 60-72h AEL, indicating a delay of 18h (**Fig 1A and 1B**). The delay is followed by an inability to transition to 3^{rd} instar (**Fig 1C**). *STIM^{KO}* larvae also exhibit retarded growth. At 72h they appear similar to CS larvae of 60h (**Fig 1D**). After 72h however, there is a complete cessation of growth in *STIM^{KO}* larvae (**Fig 1D and 1E**), followed by gradual loss of viability after 80-86h (**S1B Fig**). From these results, it became evident that cessation of growth precedes loss of viability in *STIM^{KO}* larvae.

The momentum of larval growth is maintained primarily by cell growth in the endoreplication tissues [23,24]. In a few organs like the brain and the imaginal discs growth is accompanied by constant cell division. Normally, at the end of embryonic development, mitotic cells such as a majority of neuroblasts (NBs) and imaginal disc cells enter a quiescent state [4,25,26]. Postembryonic larval development is initiated in the late first instar and early second instar stages by cell growth and renewed cell proliferation in the brain and imaginal discs, where it is nutrient dependent [26]. Cessation of growth in *STIM^{KO}* larvae (**Fig 1D and 1E**), suggested a deficit in cell growth and/or cell division. To investigate the status of cell proliferation in *STIM^{KO}* larvae we chose the well-characterized system of thoracic neuroblasts [27]. Upon comparison of thoracic segments of WT and *STIM^{KO}* larvae at 70-74h (**Fig 1F,** first two columns) it was evident that NBs exited from quiescence and entered the proliferative state in both genotypes. Both the NB marker Deadpan (red) and the post-mitotic cell marker Prospero (blue) appeared normal in *STIM^{KO}* larvae of 70-74h AEL. Subsequently, at 82-86h, the number of postmitotic cells (Prospero positive) decreased significantly in *STIM^{KO}* animals as compared to the controls but the number of thoracic neuroblasts remained unchanged (**Fig 1F,** compare third and fourth columns). Upon quantification, the ratio of dividing neuroblasts (Deadpan surrounded by Prospero positive cells) to non-dividing neuroblasts (Deadpan with either no or few Prospero positive cells) changed significantly in 86h aged *STIM^{KO}* larvae (**Fig 1G and 1H**). To identify the cause underlying the reduced number of postmitotic cells we analyzed different phases of the cell cycle of thoracic neuroblasts in *STIM^{KO}* larval brains. For this, we used the genetically encoded FUCCI system [28]. Here, the G1, S, and G2 phases of interphase are marked by green, red and green+red (yellow) fluorescent tags respectively. At 72-76h, both control and *STIM^{KO}* showed an asynchronous pattern of division. But at 82-86h, control larvae persisted with the asynchronous pattern, whereas a majority of thoracic neuroblasts in *STIM^{KO}* animals remained in the G2/M state (**S1C Fig**).

Among other factors, the exit of quiescence and maintenance of neuroblast proliferation at the early second instar stage depends on nutritional intake [26,29,30]. The slow growth and

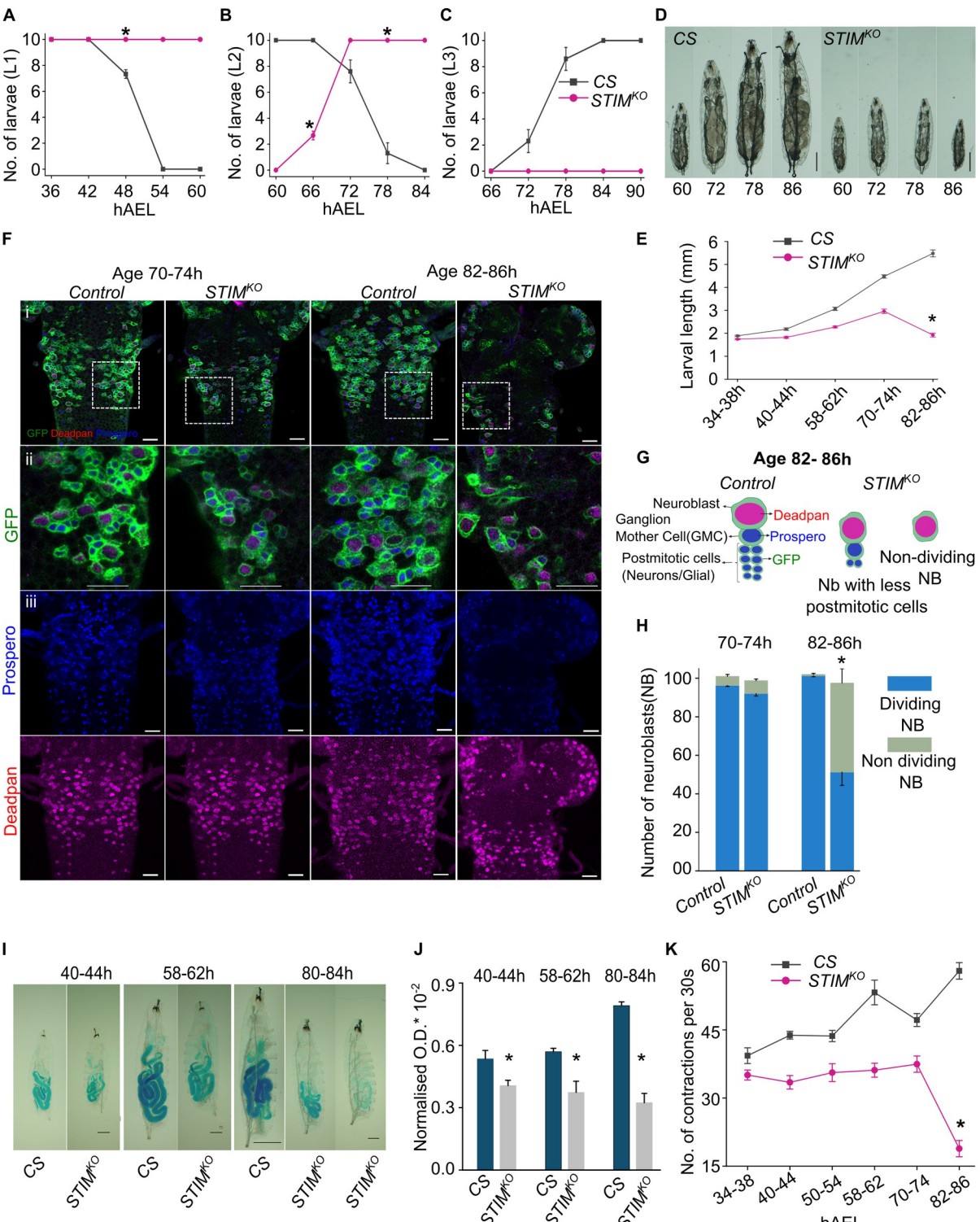

**Fig 1. $STIM^{KO}$ larvae exhibit reduced feeding followed by systemic growth arrest. (A-C)** Number of 1st instar (L1), 2nd instar (L2) and 3rd instar (L3) larvae from $CS$ (grey) and $STIM^{KO}$ (magenta) measured at 6h intervals after egg laying (AEL) at the specified time points (mean ± SEM). Number of sets (N) = 3, number of larvae per replicate (n) = 10. *P < 0.05, Student's *t*-test with unequal variances. P values are given in **S2 Table**. **(D)** Representative images of larvae from $CS$ and $STIM^{KO}$ at the indicated time. Scale bar = 1 mm. **(E)** Measurement of larval length (mean ± SEM) from $CS$ (grey) and $STIM^{KO}$ (magenta) larvae at the specified time points. Number of larvae per genotype per time point is (n) ≥ 12. *P < 0.05 for all time points, Student's *t*-test with unequal variances. P values are given in **S2 Table**. **(F)** Representative images of thoracic neuroblasts marked with $Insc>mCD8GFP$ (green), a neuroblast marker (anti-Deadpan, red) and a marker for post-mitotic cells (anti-Prospero, blue) from control ($Insc>mCD8GFP$) and $STIM^{KO}$; $Insc>mCD8GFP$ animals at the indicated ages. Similar images were obtained from four or more animals. Scale bar = 20μm. **(G)** Diagrammatic summary of neuroblast proliferation in control ($Insc>mCD8GFP$) and $STIM^{KO}$; $Insc>mCD8GFP$ animals. **(H)** Stack bar graph showing number of dividing neuroblasts to non-dividing neuroblasts. *P < 0.05, Student's *t*-test with unequal variances, n = 4 animals from each genotype. P values are given in **S2 Table**. **(I)** Representative images of dye-fed larvae from $CS$ and $STIM^{KO}$ at the indicated times AEL, scale bar = 200μm except for $CS$ (80-84h) where scale = 1mm. **(J)** Quantification (mean ± SEM) of ingested blue dye in $CS$ and $STIM^{KO}$ larvae at the indicated ages by normalizing optical density (OD) of the dye at 655nm to concentration of protein. Number of feeding plates per time point (N) = 6, number of larvae per plate (n) = 10. *P < 0.05, Student's *t*-test with unequal variances. P values are given in **S2 Table**. **(K)** Line graph with quantification of larval mouth hook contractions per 30 seconds (mean ± SEM) from $CS$ and $STIM^{KO}$ at indicated developmental time points. Number of larvae per genotype per time point is (n) ≥ 10. *P < 0.05 at all time points, Student's *t*-test with unequal variances. All P values are given in **S2 Table**.

delayed exit from quiescence suggested that $STIM^{KO}$ larvae may lack adequate nutritional inputs. As a first step staged larvae were placed on yeast, mixed with a blue dye and tested for ingestion of food. Even as early as 40-44h AEL there was a significant reduction of food intake in $STIM^{KO}$ larvae (**Fig 1I**, quantification in **Fig 1J**). By 80-84h AEL two classes of $STIM^{KO}$ larvae were evident. One with reduced food intake and others with no food intake (**Figs 1I and S1D Fig**). The proportion of $STIM^{KO}$ larvae with no food intake reached ~70% by 82-86h AEL (**S1E Fig**). The ability to feed was further quantified in $STIM^{KO}$ animals by measuring mouth hook contractions through larval development [31]. Control larvae ($CS$) exhibit a steady increase in mouth hook contractions with age, except prior to and during larval molts, indicating greater nutrient intake with age. In contrast, increase in mouth hook movements of $STIM^{KO}$ larvae follows a slower developmental trajectory, with minimal increase as they progress from first to second instar larvae and a cessation at 74h AEL, that is further retarded at 86h (**Fig 1K**; **S1–S4 Videos**). Thus, the acceleration of mouth hook movements observed in $CS$ larvae from first to third instar is retarded in $STIM^{KO}$ larvae before the appearance of growth deficits (**Fig 1I–1K**) suggesting that the consequent nutritional deficits prevent normal growth.

## STIM function is required in a subset of larval dopaminergic neurons

To understand how the loss of STIM might affect larval feeding we identified specific cells that require STIM function for larval growth and viability. From previous work, we know that knock out of $STIM$ in dopaminergic neurons marked by $THGAL4$ [32] leads to larval lethality [22]. Over-expression of a wildtype $UASSTIM^+$ transgene, henceforth referred to as $STIM^+$, in dopaminergic neurons marked by $THGAL4$ rescued larval lethality of $STIM^{KO}$ animals to a significant extent (**S2A Fig**). Absence of complete rescue by $STIM^+$ expression in dopaminergic cells suggests additional requirement for STIM in non-dopaminergic cells of $STIM^{KO}$ larvae, not investigated further in this study. Further to identify specific dopaminergic neurons that require STIM function for growth and viability we tested rescue by overexpression of $STIM^+$ in two non-overlapping subsets of dopaminergic neurons marked by $THC'GAL4$ and $THD'GAL4$ [33] and henceforth referred to as $THD'$ and $THC'$. Rescue of $STIM^{KO}$ larvae from 2nd to 3rd instar (~90%) was evident upon over-expression of $STIM^+$ in THD' marked neurons (**Fig 2A and 2B**). Because developmental profiles of the control genotypes $STIM^{KO}$; THD' and $STIM^{KO}$; $STIM^+$ are similar to $STIM^{KO}$ at 80-86h and at 168-174h (**S2A Fig**) these were not included in the experiment in Fig 2A. Though unlikely, the developmental profile of rescue larvae ($STIM^{KO}$ THD'; $STIM^+$) between 72h and 84h may thus in part be due to presence of the

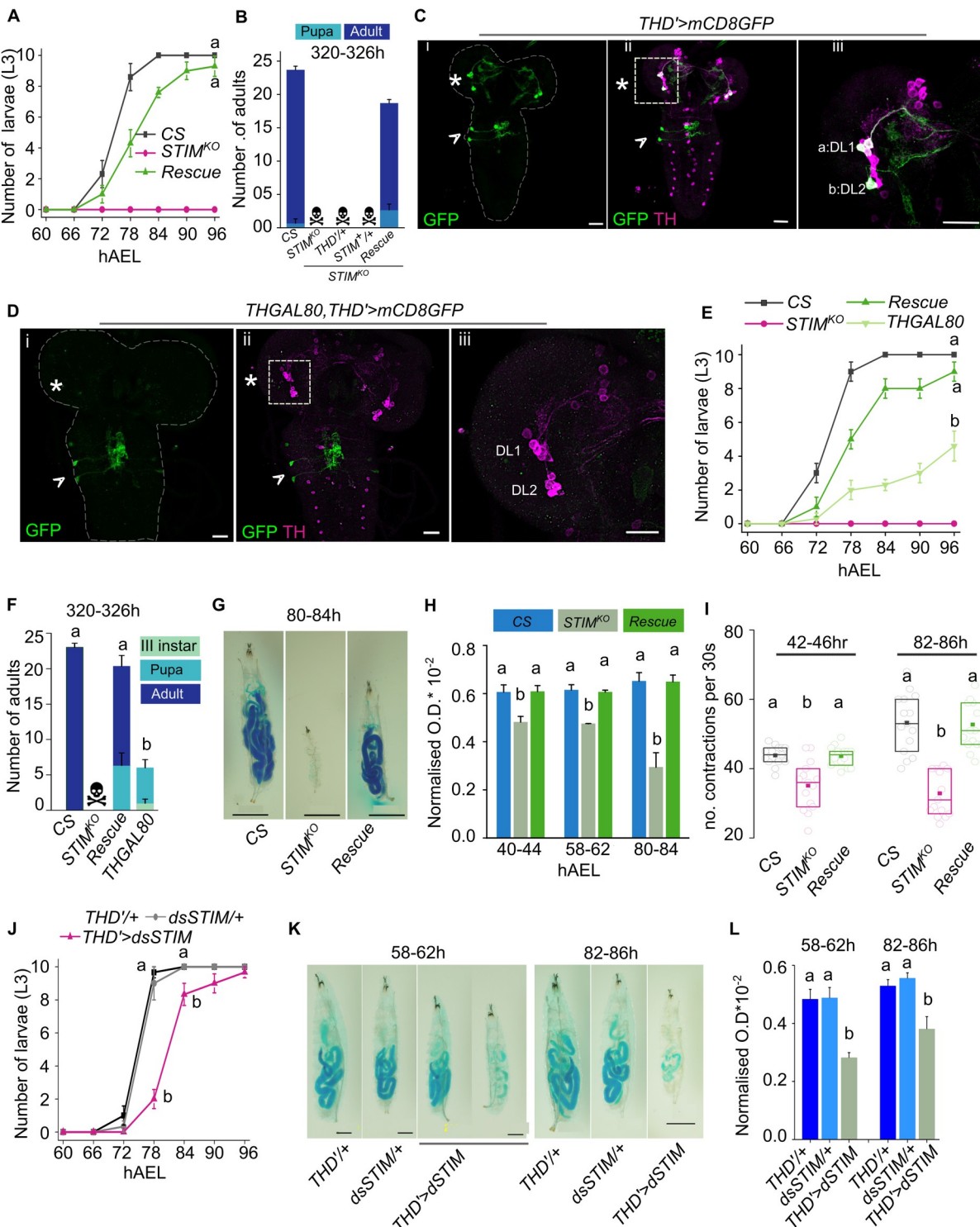

**Fig 2. Feeding and growth deficits of $STIM^{KO}$ larvae arise from central dopaminergic neurons. (A)** Number (mean ± SEM) of 3rd instar larvae (L3) are restored close to wildtype (CS) levels by expression of $STIM^+$ in THD' cells of $STIM^{KO}$ larvae (rescue). Larvae were monitored at 6h intervals from 66h to 96h AEL. Number of sets (N) = 3, number of larvae per set (n) = 10. Letters represent statistically similar groups for the 90h and 96h time point. Also see **S2A Fig** for a complete developmental profile of CS, $STIM^{KO}$; THD'>$STIM^+$) animals with appropriate genetic controls. **(B)** Stack bars with the number of adults (mean ± SEM) that eclosed at 320 to 326h AEL from the indicated genotypes. The genotype of rescue larvae is $STIM^{KO}$; THD'>$STIM^+$. Number of sets (N) = 3, number of organisms per set (n) = 25. **(C)** Representative confocal images of the larval brain from animals of the genotype THD'>mCD8GFP. Anti-GFP (green) indicates the expression of THD'GAL4 and anti-TH (magenta) marks all dopaminergic cells. Asterisks mark TH$^{+ve}$ cells in CNS whereas arrowheads mark non-TH positive cells in ventral ganglia of larval brain (i and ii). DL1 and DL2 clusters in the central brain of three and two dopaminergic cells respectively are marked (iii). Scale bars = 20µm. **(D)** Representative confocal images of the larval brain from animals of the genotype THGAL80,THD'>mCD8GFP. THD'GAL4 driven GFP expression (green) is suppressed in DL1 and DL2 clusters in the CNS (asterisk) by THGAL80 but not in the ventral ganglia (arrowheads). Scale bar = 20µm. **(E)** Line graph shows the number (mean ± SEM) of 3rd instar larvae from CS, $STIM^{KO}$, $STIM^{KO}$;THD'>$STIM^+$ and $STIM^{KO}$;THGAL80,THD'>$STIM^+$ at 6h intervals between 60 to 96h AEL. Number of sets (N) = 3, number of larvae per set (n) = 10. Different alphabet represent statistically significant groups for 84h, 90h and 96h. Also see **S2A Fig** for complete larval staging profile and additional genetic controls. **(F)** Stack bar graph showing the number of adults eclosed (mean ± SEM) at 320 to 326h AEL from CS (wildtype), $STIM^{KO}$ (mutant), $STIM^{KO}$;THD'>$STIM^+$ (rescue) and $STIM^{KO}$;THGAL80,THD'>$STIM^+$ (THGAL80—restrictive rescue) genotypes. Different alphabet represent statistically significant groups. Number of sets (N) = 3, number of organisms per set (n) = 25. **(G)** Representative images of dye-fed larvae of CS, $STIM^{KO}$ and rescue ($STIM^{KO}$; THD'>$STIM^+$) genotypes at 80-84h AEL Scale bar = 1mm. **(H)** Bar graph with quantification of ingested food containing a blue dye in larvae of the indicated genotypes (CS, $STIM^{KO}$ and $STIM^{KO}$; THD'>$STIM^+$ rescue) at the indicated developmental times. Mean (mean ± SEM) optical density (655nM) of blue dye in larval lysates after normalizing to larval protein concentration (OD/Protein concentration X 10$^{-2}$) was obtained from 6 feeding plates (N) each containing 10 larvae (n). Different alphabet represent statistically significant groups. **(I)** Expression of $STIM^+$ in THD' cells rescues the feeding behaviour deficit of $STIM^{KO}$ larvae. Box graph with quantification of larval mouth hook contractions of the indicated genotypes (CS, $STIM^{KO}$ and $STIM^{KO}$; THD'>$STIM^+$ rescue). Circles represent single larvae in the box graph of 25th and 75th percentiles with the median (bar), and mean (square). Number of larvae per genotype per time point is (n) ≥10. Different alphabet represent statistically significant groups. **(J)** Number of 3rd instar larvae (mean ± SEM) from RNAi knockdown of $STIM^+$ in THD' neurons (THD'>dsSTIM) along with control genotypes THD'/+ and dsSTIM/+ at 6h intervals between 66h to 96h AEL (After Egg laying). Number of sets (N) = 3, number of larvae per set (n) = 10. Also see **S2B** and **S2C Fig** for complete larval staging profile and adult weights. Different alphabet represent statistically significant groups. **(K)** Representative images of dye-fed larvae of the indicated genotypes at 58-62h and 82-86h AEL. Scale bar = 200µm. **(L)** Quantification (mean ± SEM) of blue dye containing ingested food (OD/Protein concentration X 10$^{-2}$; similar to panel D above) in larvae of the indicated genotypes at 58-62h and 82-86h AEL. No. of plates for each time point, (N) = 3, number of larvae per plate (n) = 10. In all graphs and box plots, different alphabets represent distinct statistical groups as calculated by one way ANOVA followed by post-hoc Tukey's test. P values for individual panels are given in **S2 Table**.

THD' and STIM$^+$ transgenes on their own. The rescue by THD' was equivalent to the rescue by THGAL4 (20±2 and 18±1.5 viable adults eclosed respectively from batches of 25 larvae; **S2A Fig**; dark and light green arrows). In contrast rescue by expression of STIM$^+$ in THC'GAL4 marked neurons was considerably less (out of batches of 25 animals 5±1 adults eclosed; **S2A Fig**; blue arrow), indicating a greater requirement for STIM function in THD' marked dopaminergic neurons.

Next, we analysed THD' driven expression of mCD8GFP and identified two classes of GFP positive cells in the larval brain. All GFP expressing cells in the central brain (three cells of DL1 and two cells of DL2 clusters [11,34,35], appear positive for Tyrosine Hydroxylase (TH) but a pair of THD' cells in the ventral ganglion (VG) appear TH negative (TH$^{-ve}$) (**Fig 2C**). To understand the relative contribution of the VG-localised TH$^{-ve}$ cells to THD'>STIM$^+$ rescue of $STIM^{KO}$ animals, we restricted THD'GAL4 expression to VG localised THD' neurons using THGAL80 [36] (**Fig 2D**). Expression of STIM$^+$ in the VG localised THD' neurons (THD'GAL4, THGAL80) reduced the rescue of $STIM^{KO}$ larvae significantly (**Figs S2A and 2E**) and was absent in adults (**Fig 2F**). Thus, the rescue of viability in $STIM^{KO}$ animals derives to a significant extent from brain-specific THD' dopaminergic neurons.

Ingestion of food and frequency of mouth hook contractions is also rescued by THD'>STIM$^+$ expression in $STIM^{KO}$ larvae (**Fig 2G–2I**). To further confirm the relevance of STIM function in THD' marked dopaminergic neurons we knocked down STIM using a previously characterized STIM RNAi (dsSTIM) [37]. THD'>dsSTIM animals exhibit delayed larval growth (**Figs 2J and S2B**) and reduced feeding (**Fig 2K and 2L**) but no larval lethality. Adults however exhibit reduced body weight (**S2C Fig**). Taken together these data identified a CNS-specific subgroup of dopaminergic neurons that require STIM function for persistent feeding during the early stages of larval growth.

## Neuronal excitability and dopamine release requires STIM

The status of THD' marked central brain dopaminergic cell clusters, DL1 and DL2 was investigated next in $STIM^{KO}$ larval brains at 80-84h, when a few viable organisms are still present despite cessation of growth and feeding (**Fig 2G**). THD' cells were marked with GFP in controls, $STIM^{KO}$ and $STIM^{KO}$ animals with $STIM^+$ rescue and the brains were stained with anti-TH sera. $THD'>mCD8GFP$ cells appeared no different in $STIM^{KO}$ as well as $STIM^+$ rescued $STIM^{KO}$ animals at 80-84h AEL when compared to controls at either 58-62h or 80-84h AEL (**S3A Fig**). Moreover, the numbers of THD' GFP cells and TH positive cells in the CNS also appeared identical (**S3B Fig**). Therefore, the loss of STIM does not lead to the loss of dopaminergic neurons in the larval brain.

In order to test if reduced feeding in $STIM^{KO}$ larvae is indeed due to a loss in dopamine signalling we measured larval feeding with knockdown of a key dopamine synthesising enzyme Tyrosine Hydroxylase (TH), in THD' neurons ($THD'>dsTH$). Knockdown of $TH$ led to significantly fewer mouth hook contractions in larvae at 80-86h AEL (**Fig 3A**, **S5–S7 Videos**), indicating reduced feeding, This was accompanied by slower progression through larval moults and some mortality at each larval stage. Finally out of a total of 25 just 20±1.2 3<sup>rd</sup> instar larvae pupated, of which 15 adults emerged (**S3C Fig**). In agreement with lower nutrient intake during larval stages, third instar larvae were smaller in size (**Fig 3B–3C**), and gave rise to adults with significantly reduced body weight (**Fig 3D**).

To understand how loss of STIM in THD' marked neurons might affect their neuronal function, we investigated properties of excitation. For this purpose, Potassium Chloride (KCl, 70mM) evoked cytosolic calcium transients were measured from THD' neurons using the genetically encoded $Ca^{2+}$ sensor GCaMP6m in *ex vivo* preparations of similarly staged control (58-62h AEL) and $STIM^{KO}$ (70-74h AEL) larvae. We chose these time points because at 72h $STIM^{KO}$ larvae appear healthy and developmentally similar to control larvae at 60h (**Fig 1D**). THD' cells responded with similar changes in GCaMP intensity, in control and $STIM^{KO}$ larvae at these time points (**Fig 3E–3G**). However, the ability to evoke and maintain cytosolic $Ca^{2+}$ transients upon KCl depolarisation was lost in THD' neurons from the DL1 and DL2 clusters of $STIM^{KO}$ larvae at 76-80h AEL (**Fig 3E–3G**). Overexpression of $STIM^+$ in THD' cells of $STIM^{KO}$ larvae rescued the KCl evoked $Ca^{2+}$ response in larvae as late as 80-84h AEL (**Fig 3E–3G**).

STIM requirement for maintaining excitability of THD' neurons was tested further by KCl stimulation of THD' neurons with STIM knockdown ($THD'>dsSTIM$) from 2<sup>nd</sup> instar larvae aged 58-62h. Two classes of responses to depolarisation by KCl were observed. Most cells (70%) responded with normal or greater changes in intensity as compared to control cells, whereas in 30% of cells KCl did not evoke a $Ca^{2+}$ transient (**S3D–S3F Fig**). We attribute this heterogeneous response to differential STIM knockdown by the RNAi in individual THD' cells and a potential effect of STIM knock-down on ER-$Ca^{2+}$ homeostasis (see below).

These data suggest that loss of STIM affects membrane excitability properties and the ability of THD' neurons to respond to stimuli. This idea was tested directly by the expression of transgenes that alter membrane potential. Over-expression of an inward rectifier $K^+$ channel, Kir2.1 in THD' neurons, that is known to hyperpolarise the plasma membrane [38], resulted in developmental delays followed by the lethality of second and third instar larvae (**S4A Fig**). Further, overexpression of a bacterial $Na^+$ channel NaChBac [39] in THD' neurons of $STIM^{KO}$ larvae evinced a weak rescue of developmental deficits, including the transition to third instar larvae (4.4±0.4) and adult viability (2.4±0.6 from of batches of 15 animals; **Fig 3H–3K**). Though weak, NaChBac's rescue was consistent. We attribute the variability to a stochastic effect of NaChBac in THD' cells. This is also evident from the variable extent of rescue in growth observed in $STIM^{KO}$; $THD'>NaChBac$ larvae (**Fig 3I and 3J**). Alternatively, in addition

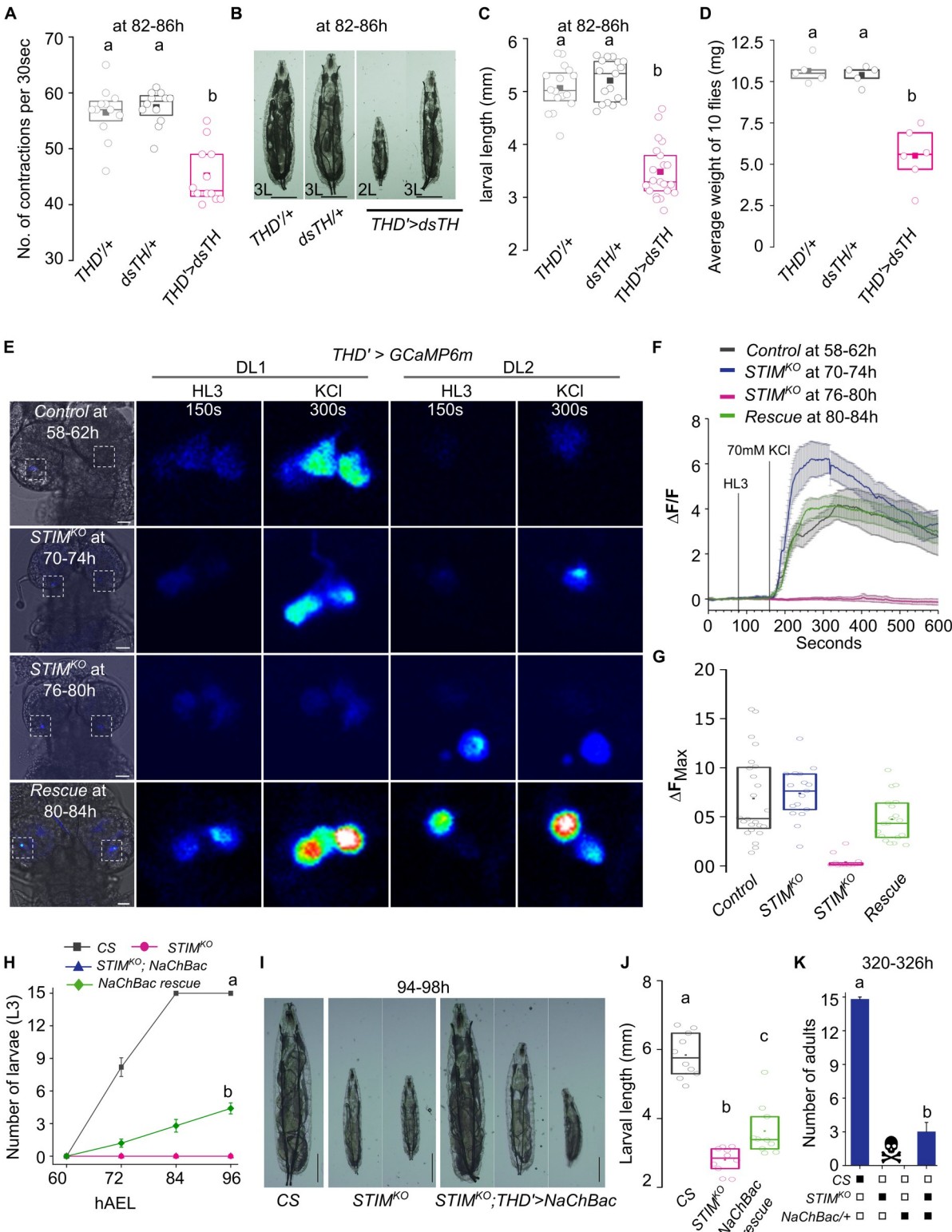

**Fig 3. Dopamine synthesis and excitability in THD' neurons is important for larval growth. (A)** Larval mouth hook movements that correlate with feeding are reduced in larvae with reduced dopamine synthesis in THD' neurons. Box graph with quantification of mouth hook contractions in larvae with knockdown of Tyrosine Hydroxylase (*dsTH*) in THD' neurons and appropriate control genotypes. Number of larvae per genotype per time point is (n) ≥10. **(B)** Representative images of larvae with knockdown of Tyrosine Hydroxylase (*THD'>dsTH*) and controls (*THD'/+*, *dsTH/+*) at 82-86h. Scale bar = 1mm. **(C)** Quantification of larval length from the indicated genotypes. n≥15. **(D)** Quantification of weight of 10 flies from the indicated genotypes. Each circle represents one set of adult flies consisting of 5 females and 5 males, 6-8h post-eclosion. A minimum of 5 sets were measured for each genotype. **(E)** Representative images of the central brain (left panels) indicating the region of focus (boxed), followed by images of DL1 and DL2 clusters of THD' cells from two lobes of same brain with $Ca^{2+}$ transients at the indicated time points before and after addition of a depolarizing agent (KCl– 70mM). $Ca^{2+}$ transients were measured by changes in the intensity of GCaMP6m fluorescence from *THD'>GCaMP6m* (control), *STIM^{KO}; THD'>GCaMP6m*, *STIM^{KO};THD'>GCaMP6m, STIM^+* (rescue). Scale bar 20μm. **(F)** Changes in GCaMP6m fluorescence (mean ± SEM of ΔF/F) from THD' neurons of the indicated genotypes. Number of brains, (N) ≥ 5, number of cells, (n) ≥ 15. **(G)** Peak intensities of GCaMP6m fluorescence (ΔF) in THD' cells from the indicated genotypes. Box plots show 25th and 75th percentiles, the median (bar), and mean (square) of ΔF of each cell (small circles). **(H)** Number of 3rd instar larvae of *CS*, *STIM^{KO}* and *STIM^{KO};THD'>NaChBac* genotypes at the indicated times AEL (mean ± SEM). Number of sets (N) = 3, number of larvae per set (n) = 15. **(I)** Representative images of larvae from *CS*, *STIM^{KO}* and *STIM^{KO}; THD'>NaChBac* at 94-98h AEL, scale bar = 1mm. **(J)** Quantification of larval length from the indicated genotypes. n = 10. **(K)** Number of adults eclosed at 320h AEL (mean ± SEM) from wildtype (*CS*), mutant (*STIM^{KO}*) and NaChBac rescue (*STIM^{KO}; THD'>NaChBac*) animals. Numbers were obtained from three experiments (N = 3), number of organisms per experiment, n = 15. In all box plots, circles represent single larvae or flies. The box plots span 25th and 75th percentiles with the median (bar), and mean (square). Alphabets represent distinct statistical groups as calculated by one way ANOVA followed by post-hoc Tukey's test. P values are given in **S2 Table**.

to neuronal excitability, STIM might affect other cellular functions such as ER stress, that are not alleviated by expression of NaChBac, resulting in the weak rescue. Control animals with overexpression of *NaChBac* in THD' neurons exhibit delayed pupariation (**S4B Fig**).

Neuronal excitability is required for neurotransmitter release at presynaptic terminals. We hypothesized that dopamine release from THD' neurons might be affected in *STIM^{KO}* larvae. To test this idea, we identified the pre-synaptic (green) and post-synaptic (red) terminal regions of THD' neurons by marking them with *Syt-eGFP* and *Denmark* respectively (**Fig 4A**) [40]. Analysis of pre-synaptic regions (Syt-eGFP expression) identified three distinct areas in the CNS. One at the centre of the CNS corresponding to the mushroom body (**Fig 4A**; asterisk), the second as a branched form in the basomedial protocerebrum of the CNS (**Fig 4A**; arrowhead) and the third one consisting of punctae spread across the oesophageal regions where brain-gut interactions take place (**Fig 4A**; hash). Based on the projection patterns observed we speculate that cells marked by THD' correspond to DL1-2, DL1-5, DL1-6 from the DL1 cluster and DL2-2 and DL2-3 from the DL2 cluster [35].

Next, dopamine release was measured in the most prominent presynaptic areas of THD' neurons, corresponding to the MB and the basomedial protocerebrum, by change in fluorescence of the genetically encoded fluorescent GPCR-activation-based-DopAmine sensor (GRAB$_{DA}$) [41]. Dopamine release in THD' neurons of *STIM^{KO}* larvae at 76-80h is significantly attenuated as compared with controls (**Fig 4B and 4C**). Importantly, overexpression of STIM^+ in THD' neurons rescued dopamine release, though with altered dynamics from control animals (**Fig 4B and 4C**; see discussion). We chose to measure dopamine release in 76-80h *STIM^{KO}* larvae because THD' neurons in their brains no longer responded to KCl evoked depolarization (**Fig 3B**) even though the larvae appear normal (**Fig 1D**). Dopamine release was stimulated by Carbachol (CCh), an agonist for the muscarinic acetylcholine receptor (mAChR), that links to $Ca^{2+}$ release from ER-stores through the ER-localised IP$_3$ receptor [42] and is expressed on THD' neurons [43]. CCh-induced changes in ER-$Ca^{2+}$ were tested directly by introducing an ER-$Ca^{2+}$ sensor [44] in THD' neurons (**Fig 4D and 4E**). Though ER-$Ca^{2+}$ release, in response to CCh could be measured in just 7 out of 23 cells, the subsequent step of ER-store refilling, presumably after Store-operated $Ca^{2+}$ entry into the cytosol through the STIM/Orai pathway, could be observed in all control THD' cells (58-62h AEL), whereas it was absent in THD' neurons from *STIM^{KO}* brains (76-80h AEL; **Fig 4E**). The ER-$Ca^{2+}$ response was rescued by over-expressing STIM^+ in THD' neurons (**Fig 4D, and 4E**). Taken together our data establish an important role for STIM-dependent ER-$Ca^{2+}$ homeostasis in maintaining

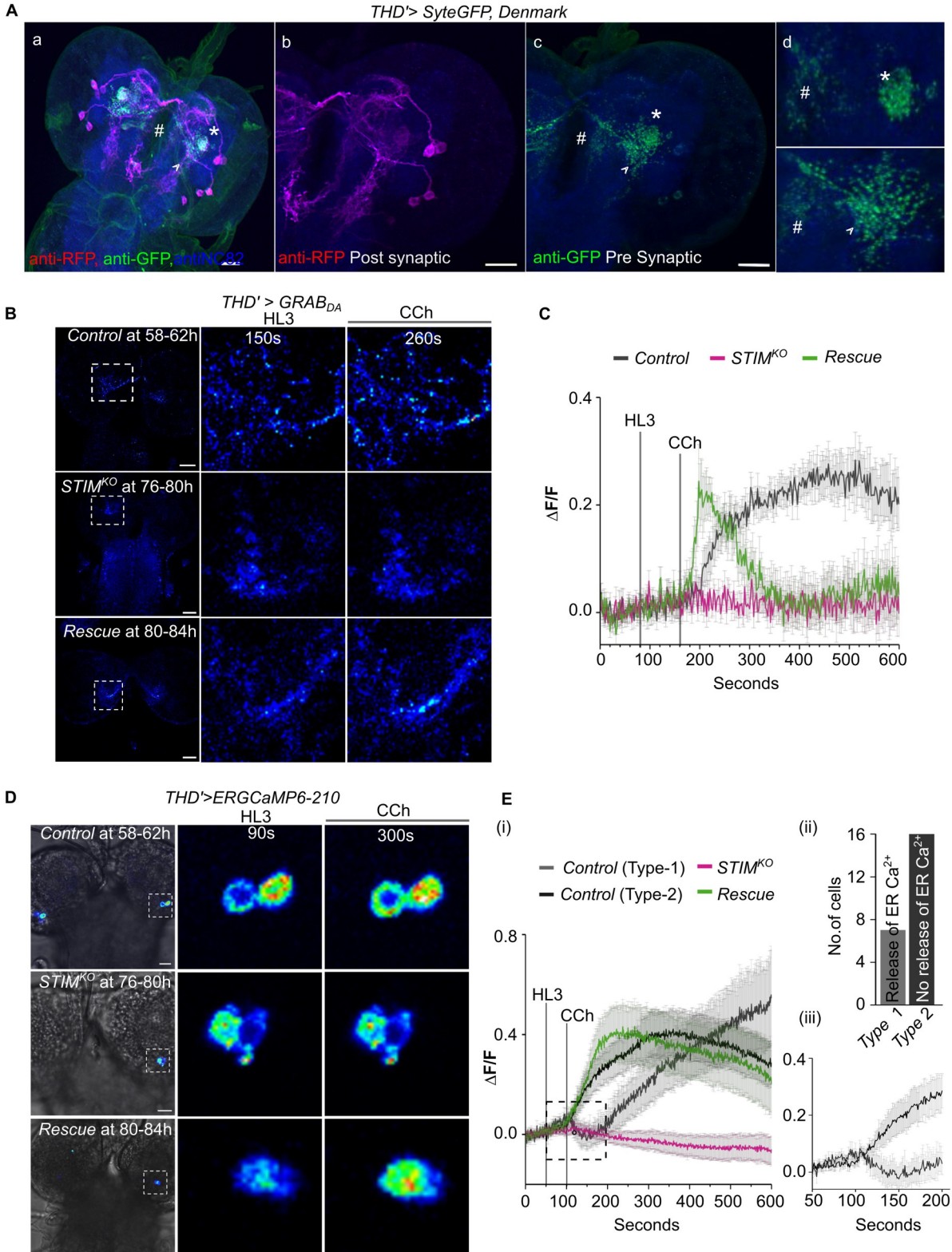

**Fig 4. STIM is necessary for dopamine release and ER-Ca$^{2+}$ homeostasis. (A)** Axonal and dendritic projections of THD' neurons visualized by expression of SyteGFP (green) and Denmark (anti-RFP, magenta) respectively in a representative brain immunostained for anti-Brp (blue). Panels marked as (d) contain magnified images from (c) of presynaptic terminals (green) located at the base of the mushroom body in the CNS (asterisk), as branches extruding into basomedial protocerebrum of the CNS (arrowhead) and as punctae near the oesophageal region (hash). **(B)** Representative images of dopamine release before and after addition of Carbachol (CCh) as measured by changes in intensity of GRAB$_{DA}$ at the presynaptic terminals of THD' neurons of control (*THD'>GRAB$_{DA}$*), *STIM$^{KO}$* (*STIM$^{KO}$;THD'>GRAB$_{DA}$*) and rescue (*STIM$^{KO}$;THD'>GRAB$_{DA}$, STIM$^+$*) genotypes. Scale bar = 20μm. **(C)** Normalized changes in fluorescence (ΔF/F) of GRAB$_{DA}$ measuring dopamine release from THD' neurons of control (*THD'>GRAB$_{DA}$*), *STIM$^{KO}$* (*STIM$^{KO}$;THD'>GRAB$_{DA}$*) and rescue (*STIM$^{KO}$;THD'>GRAB$_{DA}$, STIM$^+$*) genotypes. Traces show the average change in GRAB$_{DA}$ florescence (mean ± SEM of (ΔF/F)) measured from individual presynaptic regions of interest (≥10) taken from N ≥ 6 brains of each genotype. **(D)** Representative time series images of ER calcium transients as measured by changes in intensity of ER-GCaMP in THD' neurons of control (*THD'>ER-GCaMP-210*), *STIM$^{KO}$* (*STIM$^{KO}$;THD'> ER-GCaMP-210)* and rescue (*STIM$^{KO}$;THD'> ER-GCaMP-210, STIM$^+$*) genotypes. **(E)** i) Traces of normalized ER-GCaMP (ΔF/F) responses (mean ± SEM) from THD' neurons of the indicated genotypes. Control (*THD'>ER-GCaMP-210*) Type 1, indicates cells that exhibit ER-Ca$^{2+}$ release upon CCh addition followed by refilling of ER-stores. Control Type 2 indicates cells where ER-Ca$^{2+}$ release was not evident. Rescue indicates *STIM$^{KO}$; THD'>ERGCaMP-210,STIM$^+$*. For each genotype, number of cells (n) ≥ 15 and N ≥ 5 brains. (ii) Quantification of control cells that exhibit ER Ca$^{2+}$ release (Type 1) and cells where ER Ca$^{2+}$ release was not observed (Type 2). (iii) Boxed region from (i) enlarged to show ER Ca$^{2+}$ response of Type 1 and Type 2 control cells.

optimal dopamine release from THD' neurons in turn required for the larval drive to feed constantly.

Because late third instar larvae stop feeding [3,19,20,45] we hypothesized that CCh-stimulated Ca$^{2+}$ responses in THD' cells might change in wandering stage third instar larvae. To test this idea we monitored carbachol-stimulated GCaMP activity in THD' neurons from wild-type larvae at 96h AEL (mid 3$^{rd}$ instar), 120h AEL (early wandering stage) and 124h AEL (late wandering stage). A robust GCaMP response was observed at 96h, whereas at the beginning of the wandering stage (118-122h AEL), the peak response was both reduced and delayed. A further delay in the peak response time was observed in late wandering stage larvae (122-126h AEL) (**S4E and S4F Fig**). Thus with gradual cessation of feeding in late third instar larvae, the dynamics of CCh-stimulated Ca$^{2+}$ responses in THD' neurons also undergo changes (see discussion).

## THD' dopaminergic neurons both activate and inhibit specific neuropeptidergic cells

A role for neuropeptides in modulating larval feeding has been identified previously [5–7] Based on these findings we postulated that dopamine release from THD' cells might modulate neuropeptidergic cells in the larval brain. This idea was tested by optogenetic stimulation of red-shifted Channelrhodopsin (*CsChrimson*) [46] expressing in THD' (*THD'>UAS-Chrimson*) cells followed by Ca$^{2+}$ imaging from GCaMP6f expressing neuropeptide cells *(C929Lex-A>LexAopGCaMP6f)* (**Fig 5A**). Upon optogenetic activation of THD' neurons, a change in cellular Ca$^{2+}$ signals was observed in a total of 64 peptidergic neurons from 9 brains, including some lateral neurosecretory cells (LNCs), median neurosecretory cells (MNSc), and regions of the suboesophageal zone (SEZ) (**Fig 5B and 5C**). Elevated Ca$^{2+}$ signals were observed in a subset of neuropeptidergic cells (**Fig 5B–5D,** yellow asterisks, n = 24), whereas in some cells Ca$^{2+}$ signals were reduced (**Fig 5B, 5C and 5E,** red asterisks, n = 40). There was no change in cellular Ca$^{2+}$ upon optogenetic stimulation in the remaining cells with visible GCaMP6f fluorescence (n = 35). Upon cessation of optogenetic activation of THD' neurons Ca$^{2+}$ levels returned to baseline (**Fig 5D and 5E**). Individual cells exhibit either activatory or inhibitory responses upon repeated optogenetic stimulation with pulses of red light (**Fig 5F (i) and (ii)**), indicating that THD' cells evoke specific responses of either stimulation or inhibition based on the class of neuropeptide cells. Specificity of the *THD'>Chrimson* evoked response was further confirmed by testing brains from larvae that were reared without all *trans*-Retinal (ATR, a cofactor for the function of Channelrhodopsin) [47] and by imaging in the absence of light (**Fig 5G**).

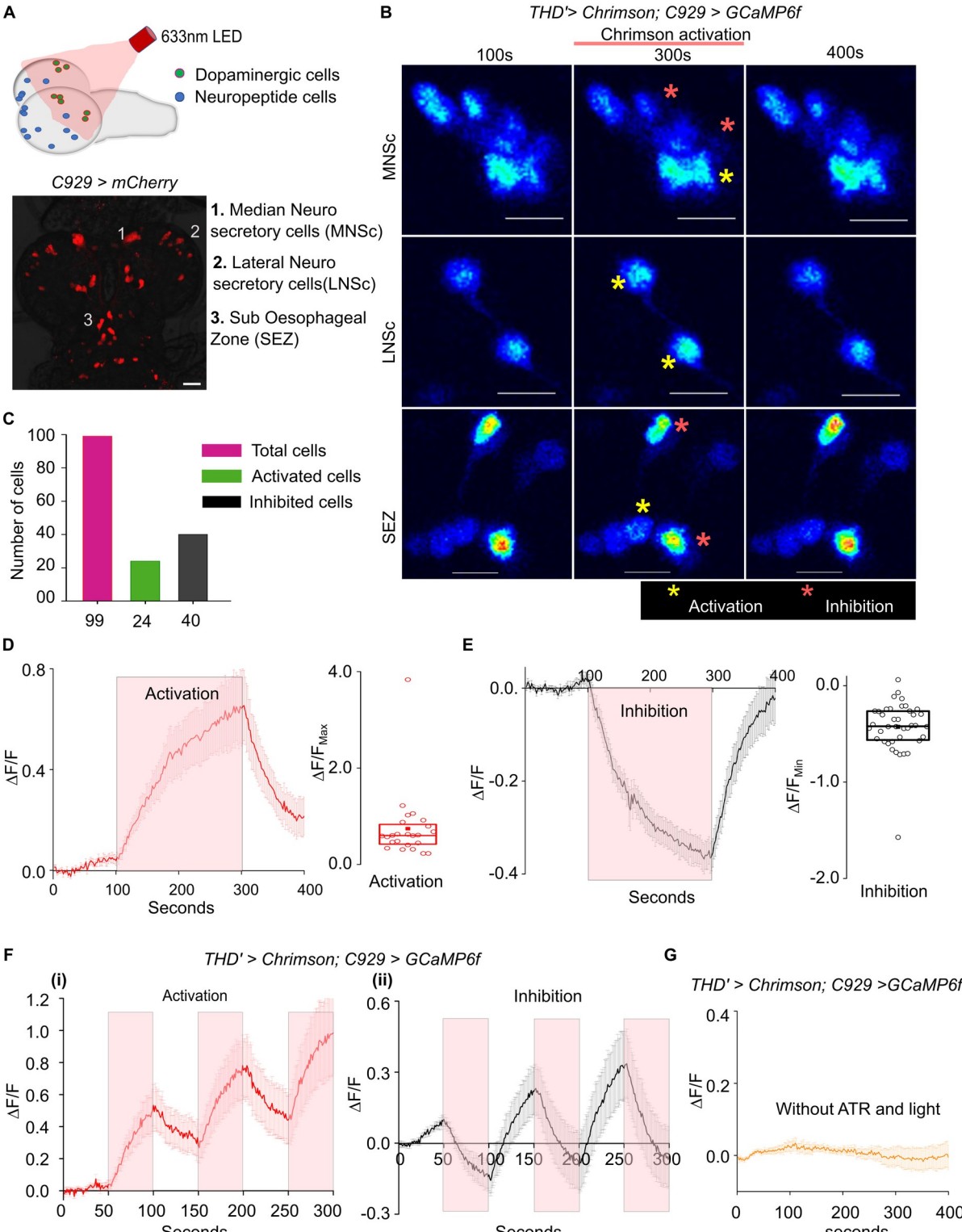

**Fig 5. Differential regulation of neuropeptidergic cell activity by THD' neurons.** (A). Pictorial representation of an experimental setup for testing functional connectivity of dopaminergic (THD') and peptidergic (C929) cells in organisms of the genotype *THD'>Chrimson, C929LexA>LexAopGCaMP6m*. The lower panel shows the peptidergic neurons containing regions 1–3 in which change in $Ca^{2+}$ transients were observed upon activation of THD' cells. (B). $Ca^{2+}$ transients in peptidergic Median Neurosecretory cells (MNSc, top), lateral neurosecretory cells (LNSc, middle) and suboesophageal region (SEZ, bottom) as measured by the intensity of GCaMP6m before Chrimson activation (100s) and after Chrimson activation (300s and 400s) of THD' cells. Activated cells (red asterisks) and inhibited cells (yellow asterisks) are marked. Scale bar = 20um. (C). Quantification of total C929 marked cells from 9 brains that were either optogenetically activated or inhibited. (D). Average GCaMP6m responses (mean ± SEM of ΔF/F) across time in C929 marked cells that were activated upon Chrimson activation of THD' neurons at 633nm. The box plot shows 25th and 75th percentiles of peak responses measured by change in GCaMP6m of individual activated peptidergic cells (circles), with the median (bar), and mean (square). n = 24 cells from (N) = 9 brains. (E). Average GCaMP6m responses (mean ± SEM of ΔF/F) across time in C929 marked cells that were inhibited upon Chrimson activation of THD' neurons at 633nm. The box plot shows 25th and 75th percentiles of peak responses measured by change in GCaMP6m of individual inhibited peptidergic cells (circles), with the median (bar), and mean (square). n = 40 cells from (N) = 9. (F). (i) Pulsed activation of THD' neurons produce corresponding pulses of activation in certain peptidergic neurons as evident from their GCaMP6m responses (mean ± SEM). Red panels indicate the time intervals (50s each) of *THD'>Chrimson* activation. n = 21 cells from N = 11 brains. (ii) Pulsed activation of THD' neurons produce corresponding pulses of inhibition in certain peptidergic neurons as evident from their GCaMP6m responses (mean ± SEM). Red panels indicate the time intervals (50s each) of *THD'>Chrimson* activation. n = 19 cells from N = 9 brains. (G). GCaMP6m responses (mean ± SEM) are absent in C929 cells in the absence of red light and in larvae that were not fed with All-trans Retinol (ATR). n = 50 cells from N = 9 brains.

## STIM expression in THD' neurons regulates the expression of insulin-like peptides

Though optogenetic stimulation of THD' cells shows dopaminergic modulation of neuropeptidergic neurons (**Fig 5A and 5B**) it does not allow identification of specific neuropeptides that function downstream of the THD' neurons. Optogenetic activation of THD' and $Ca^{2+}$ responses in neuropeptidergic cells (marked by *C929LexA>LexAopmCherry*) helped identify three clusters of neuropeptidergic cells that are downstream of THD' neurons and include the well-characterised ilp expressing MNSc cluster [48], as a putative target of THD' neurons (**Fig 5B**). Analysis of an RNAseq experiment identified changes in gene expression in brains from second instar *STIM^{KO}* larvae (72-76h AEL) with developmentally comparable CS brains (58-62h; **S5A Fig and S3 Table**), found significant changes in expression of *ilp2*, *ilp3* and *ilp5*. Whereas *ilp2* and *ilp5* were significantly downregulated, *ilp3* was upregulated more than 5 fold (**Fig 6A**). The differential regulation of *ilp2*, *-3* and *-5* expression in *STIM^{KO}* brains was further validated by qPCRs (**Fig 6B**). Importantly, expression of *ilp3* and *ilp5* were restored back to normal in brains from *STIM^{KO}* larvae, rescued by overexpression of *STIM^{+}* in THD' neurons (**Fig 6B**). During normal larval development *ilp3* transcript levels are low in actively feeding larvae (L2 and L3, 12h) and are gradually upregulated in later stages of L3 as larvae stop feeding (**Fig 6C**; DGET [49]. Based on the up-regulation of *ilp3* in *STIM^{KO}* larvae (**Fig 6A and 6B**), we hypothesised that knock down of *ilp3* in MNSc might rescue *STIM^{KO}* larvae. Indeed, partial rescue of larval lethality in 2nd instar larvae followed by their transition to 3rd instar larvae (5 ±0.5) was observed (**Fig 6D—G**). A few of the rescued 3rd instar larvae grew to full size, and pupariated (**Fig 6E**, L3b type larvae) and some even eclosed as adults (2.3±0.3 from 3 batches of 25 animals; **Fig 6D and 6E**). The partial rescue observed by *ilp3* knockdown may be due to roles of additional dopamine-modulated neuropeptides plus the lower expression of ilp2 and ilp5 in *STIM^{KO}* animals (**Fig 6A and 6B**) because both ilps are growth signals in 2nd and 3rd instar larvae. A small proportion of *STIM^{KO}* larvae rescued by *ilp3* knockdown appear significantly larger than control larvae (**Fig 6E and 6F**, L3b), indicating that loss of ilp3 can rescue growth in some animals. This idea is further supported by knock-down of *ilp3* in MNSc from wildtype animals that resulted in slower transition to pupariation and larger pupae, and overweight adults (**Figs 6H–6J and S5A–S5B**). Conversely, over-expression of *ilp3* in MNSc resulted in delayed larval transition from L2 to L3 and smaller sized larvae (**Fig 6K–6M**) similar to delayed larval development in ilp5 knockdown animals (**S5A Fig**). However, overexpression of ilp3 had no effect on feeding as indicated by measurement of larval mouth hook movements (**Fig 6N**) even though adults were of reduced body weight (**S5B Fig**). Taken

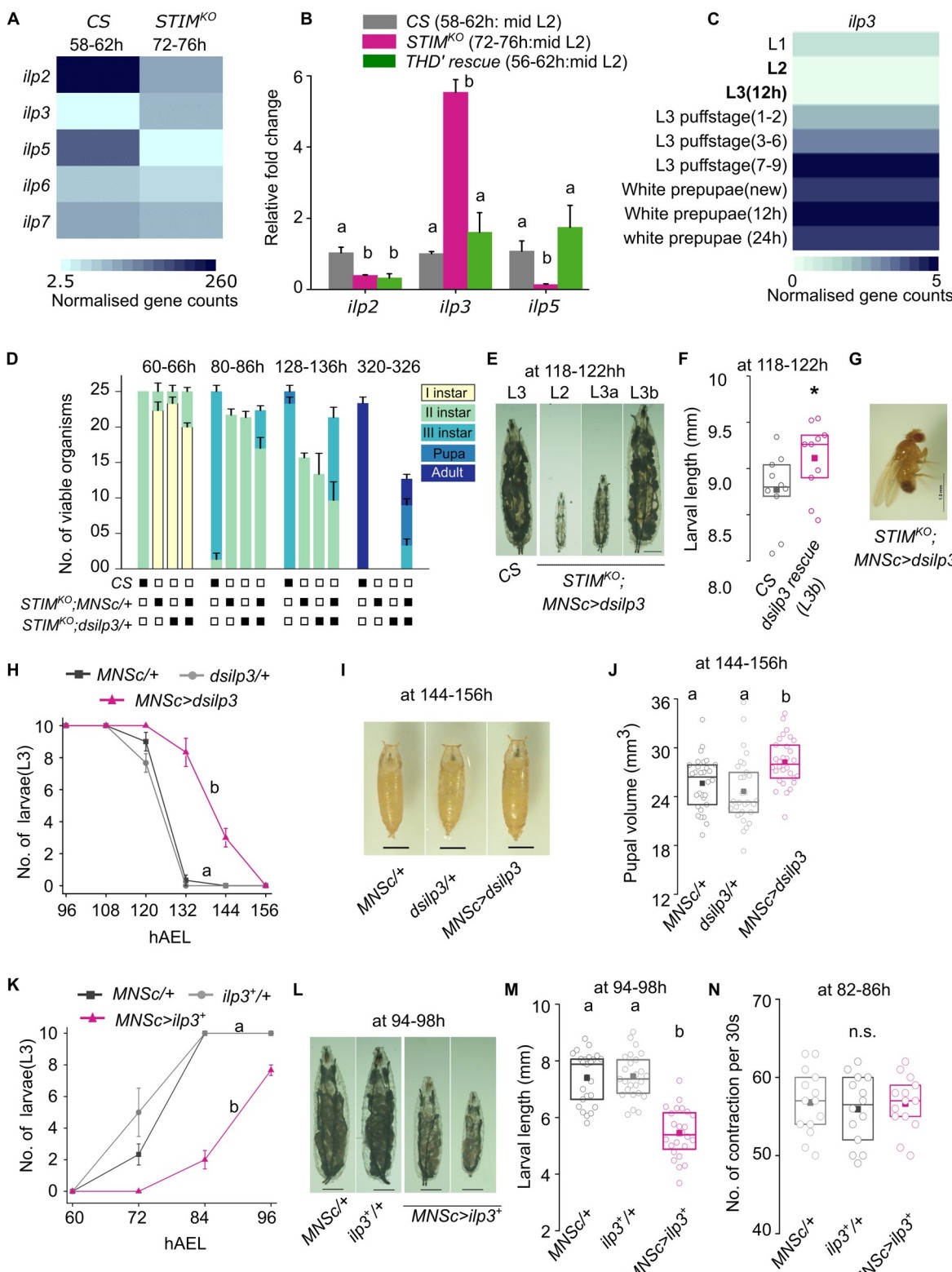

**Fig 6. Developmental expression of insulin-like peptide 3 (ilp3) is regulated by STIM in THD' neurons. (A)**. FPKM values obtained for transcripts encoding the indicated insulin like peptides (*ilp*) obtained from an RNAseq experiment performed from *CS* brains (58-62h, mid 2nd instar stage) and *STIM$^{KO}$* brains (72-76h, mid 2nd instar stage). **(B)**. Quantification of *ilp2*, *ilp3* and *ilp5* transcripts in *CS*, *STIM$^{KO}$* and *STIM$^{KO}$; THD'>STIM$^+$* (rescue) brains. Expression changes of *ilp3* and *ilp5* transcripts in *STIM$^{KO}$* brains were rescued upon over-expression of *STIM$^+$* in THD' neurons. Transcripts were measured relative to controls. Data show (mean ± SEM) of fold changes of gene expression (N = 3). **(C)**. Heatmap of *ilp3* expression from larval to pupal stages. FPKM values were obtained from the DGET database (https://www.flyrnai.org/tools/dget/web/). **(D)**. Knockdown of *ilp3* partially rescues viability in *STIM$^{KO}$* larvae. Stack bar graph showing the number of viable organisms and their developmental stage at the specified hours after egg laying for *CS*, *STIM$^{KO}$; MNScGAL4/+*, *STIM$^{KO}$; dsilp3/+* (control) genotypes followed by *STIM$^{KO}$;MNScGAL4>dsilp3*. Data are from three experiments, each with 25 organisms per genotype. **(E)**. Representative images of larvae from *CS* and *STIM$^{KO}$; MNSc>dsilp3* at 118-122h AEL, scale bar = 1mm. **(F)**. Quantification of larval length from *CS* and *STIM$^{KO}$; MNSc>dsilp3* (rescue, L3b) genotypes. Box plots show the 25th and 75th percentiles with median (bar), mean (square) and sizes of individual larvae (circles), n = 10. *P<0.05 (unpaired t-test). **(G)**. Representative image of an adult *STIM$^{KO}$* mutant fly rescued by knockdown of *ilp3* (*STIM$^{KO}$; MNScGAL4>dsilp3*). Scale bar = 1mm. **(H)**. Number of 3rd instar larvae at the indicated times AEL (mean ± SEM) from the indicated genotypes with knockdown of *ilp3* in the MNSc (*MNSc>dsilp3*) and of controls (*MNSc/+ and dsilp3/+*). Alphabets indicate statistically different groups at 132h and 144h. Number of sets (N) = 3 and number of larvae per set (n) = 10. **(I)**. Representative images of pupae with knock-down of *ilp3* (*MNSc> dsilp3*) at 144-156h AEL and appropriate controls (*MNSc/+ and dsilp3/+*). Scale bar = 1mm. **(J)**. Quantification of pupal volume from the indicated genotypes. Box plots show the 25th and 75th percentiles with median (bar), mean (square) and sizes of individual pupa (circles), n = 30. **(K)**. Number of 3rd instar larvae obtained by over-expression of *ilp3* (*MNSc>ilp3$^+$*) and in appropriate control genotypes (*MNSc/+, ilp3$^+$/+*) at the indicated time AEL (mean ± SEM). (N) = 3 and (n) = 10. Alphabets indicate statistically different groups. **(L)**. Representative images of larvae with over-expression of *ilp3* (*MNSc>ilp3$^+$*) and appropriate controls (*MNSc/+ and ilp3$^+$/+*) at 94-98h AEL, scale bar = 1mm. **(M)**. Overexpression of ilp3 results in smaller sized larvae. Quantification of larval length from the indicated genotypes. Box plots show the 25th and 75th percentiles with median (bar), mean (square) and sizes of individual larvae (circles), n ≥24. **(N)**. Overexpression of ilp3 does not affect feeding. Box graph showing 25th and 75th percentiles, the median (bar), and mean (square) of number of larval mouth hook contraction per 30 seconds, where each circle represents a single larva of the indicated genotype. Number of larvae per genotype is (n)≥10. For all graphs and box plots, alphabets represent distinct statistical groups as calculated by one way ANOVA followed by post-hoc Tukey's test, n.s. indicates non-significant groups. P values are given in S2 Table.

together, these data show that dopamine signals from THD' cells are required to maintain normal expression of growth promoting ILPs (*ilp2* and *ilp5*) and repress expression of *ilp3* that appears to function as an anti-growth signal.

A direct synaptic connection between central dopaminergic neurons and the MNSc has not been reported [5,50] and is likely not supported by data that mapped synaptic connections in the larval brain [35,50–52]. However, a neuromodulatory role for dopamine is documented where it can effect a larger subset of neurons, beyond direct synaptic partners, by means of diffusion aided volumetric transmission [53–56]. To test this possibility we began by measuring larval developmental transitions in animals with knockdown of three dopamine receptors, *Dop1R1*, *Dop2R2* and *DopEcR* in the MNSc. Among these, knockdown of *Dop1R1* delayed development, reduced larval and adult viability (**Fig 7A**) and negatively impacted growth (**Fig 7B and 7C**). A weaker phenotype of delayed development and loss of viability was observed with knockdown of DopEcR, whereas larvae with knockdown of Dop2R2 appeared normal (**Fig 7A**). The response of MNSc to dopamine was tested next. Brains expressing a $Ca^{2+}$ sensor in the MNSc (*MNSc>GCaMP6m*) were stimulated with dopamine in the presence of a $Na_v$ blocker Tetrodotoxin [57], to prevent extraneous neuronal inputs. Of the seven targeted MNSc in one hemisphere, we observed consistent activation of one cell whereas dopamine addition inhibited the $Ca^{2+}$ response in three to four cells (**Fig 7D**; quantified in **Fig 7E**). No changes in the $Ca^{2+}$ responses of MNSc were observed in the absence of dopamine (**Fig 7D and 7E**).

## Discussion

In *Drosophila*, as in other holometabolous insects, growth is restricted to the larval stages. In early stages of larval development cells exit mitotic quiescence and re-enter mitosis resulting in organismal growth [1,25,58].This change is accompanied by an increase in the feeding rate of the organism so as to provide sufficient nutrition for the accompanying growth in organismal size. In *STIM$^{KO}$* larvae we observed a loss of this ability to feed persistently starting from early second instar larvae. The focus of this feeding deficit lies in a subset of central dopaminergic neurons that require STIM function to maintain excitability. Importantly, these dopaminergic

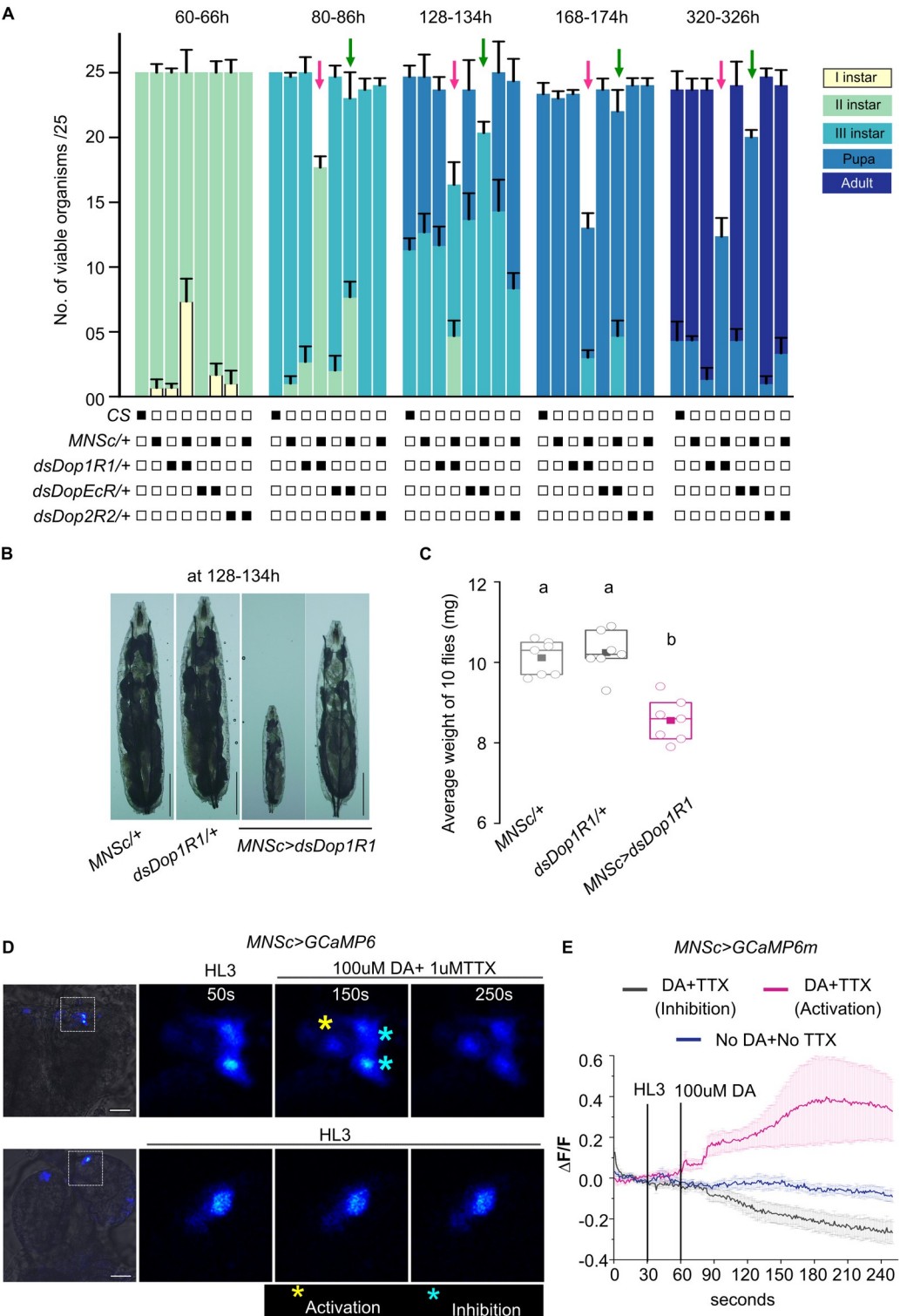

**Fig 7. MNSc respond differentially to Dopamine stimulation. (A)**. Development profiles of larvae with RNAi mediated knockdown of dopamine receptors in the MNSc. Bar graphs represent number of viable organisms and their developmental stages at the specified hours after egg laying for the indicated genotypes (*CS, MNScGAL4/+, dsDop1R1/+, dsDopEcR/+, dsDop2R1/+* (Controls), *MNSc>dsDop1R1, MNSc>dsDopEcR,* and *MNSc>dsDop2R2*). N = 3 batches of n = 25 larvae for every genotype. P values of all genotypes compared with appropriate controls were calculated by one way ANOVA followed by post-hoc Tukey's test and are given in **S2 Table. (B)**. Representative images of larvae from specified genotypes at 128-134hr. Scale bar = 1mm. **(C)**. Quantification of weight of 10 flies from the indicated genotypes. Box plot shows the 25th and 75th percentiles with median (bar) and mean (square). Each circle represents one set of adult flies consisting of 5 females and 5 males, 6-8h post-eclosion. A minimum of 5 sets were measured for each genotype. Alphabets represent distinct statistical groups as calculated by one way ANOVA followed by post-hoc Tukey's test. P values are given in **S2 Table. (D)**. Representative images of larval brains expressing the calcium indicator GCaMP6 in MNSc. Brains were stimulated with either dopamine (DA) in presence of tetrodotoxin (top panel) or a control solution (HL3; lower panel) and imaged over time. Scale bar = 20μm. **(E)**. Changes in GCaMP6m fluorescence (mean ± SEM of ΔF/F) from MNSc cells with addition of either dopamine (DA) and Tetrodotoxin (TTX) (7 brains and 12–14 cells) or HL3 (No DA+ No TTX) (5 brains and 14 cells).

neurons communicate with multiple neuropeptidergic cells in the brain (**Fig 5D**) to regulate appropriate changes in larval feeding behaviour. The identified dopaminergic cells also communicate with ilp producing neuropeptidergic cells, the MNSc, through which they appear to impact larval growth.

## STIM and excitability of dopaminergic neurons

We identified the THD' cells as critical for larval feeding from their inability to function in the absence of the SOCE regulator STIM. Loss of excitability and the absence of dopamine release from THD' cells in $STIM^{KO}$ larvae (**Figs 3A–3C and 4B–4D**) suggests that voltage-dependent receptor activity is required to maintain growth in early 2nd instar larvae. Changes in expression of ion channels and presynaptic components have been observed earlier upon knockdown of STIM in *Drosophila* and mammalian neurons [59,60]. Moreover, loss of STIM-dependent SOCE in *Drosophila* neurons effects their synaptic release properties [61]. Partial rescue of viability in $STIM^{KO}$ organisms by over-expression of a bacterial sodium channel NaChBac (**Fig 3D–3G**) and restoration of dopamine release upon rescue by $STIM^+$ (**Fig 4B and 4C**) supports the idea that STIM-dependant SOCE maybe required for appropriate function and/or expression of ion channels and synaptic components in THD' neurons. Changes in ER-$Ca^{2+}$ (**Figs 4D and 5E**) suggest that STIM is also required to maintain neuronal $Ca^{2+}$ homeostasis.

## Dopaminergic control of larval food seeking

Whilst mechanisms that regulate developmental progression of *Drosophila* larvae have been extensively studied, neural control of essential changes in feeding behaviour that need to accompany each larval developmental stage have not been identified previously. Artificial manipulation of activity in the central dopaminergic neuron subset studied here (THD'), either by expression of an inward rectifying potassium channel (Kir2.1) (**S4A Fig**) or the bacterial sodium channel NaChBac (**S4B Fig**), suggests an important role for THD' neurons during larval development. This idea is supported by the altered dynamics of muscarinic acetylcholine receptor (mAChR) stimulated $Ca^{2+}$ release observed in THD' neurons between early, mid and late third instar larvae when larval feeding slows down and ultimately stops (**S4C and S4D Fig**) and re-iterates that signaling in and from these neurons drives larval feeding whereas lower carbachol-induced $Ca^{2+}$ responses signal cessation of feeding. A weaker rescue of $STIM^{KO}$ larvae is also obtained from $STIM^+$ expression in the THC' neuron subset. Taken together these observations suggest a neuromodulatory role for dopamine, where DA release from THD' neurons has a greater influence on feeding than the DA release from THC' neurons, possibly due to the DL1 and DL2 cluster (among THD' marked neurons) receiving more

feeding and metabolic inputs [7,35,58]. A role for cells other than THD', in maintaining kinetics of dopamine release required for feeding behaviour are also indicated because expression of STIM$^+$ in THD' neurons did not revert kinetics of dopamine release to wild type levels (**Fig 4C**). The prolonged dopamine release observed in wild-type THD' neurons may arise from synaptic/modulatory inputs to THD' neurons from other neurons that require STIM function.

Though the cells that provide cholinergic inputs to THD' cells have not been identified it is possible that such neurons sense the nutritional state. In this context, two pairs of cells in the THD' subset also motivate the search for food in hungry adult *Drosophila* [62]. Starved flies with knock down of the mAChR on THD' neurons exhibit a decrease in food seeking behaviour [43]. Cholinergic inputs to THD' neurons for sensing nutritional state/hunger may thus be preserved between larval and adult stages.

Interestingly, dopamine is also required for reward-based feeding, initiation, and reinforcement of feeding behaviour in adult mice [63]. These findings parallel past studies where prenatal mice genetically deficient for dopamine (DA$^{-/-}$), were unable to feed and died from starvation. Feeding could however be initiated upon either enforced supplementation or injection with L-DOPA [64] allowing them to survive. More recent findings show that dopaminergic neurons in the ventral tegmental area (VTA), and not the substantia nigra, drive motivational behaviour and facilitate action initiation for feeding in adult mice [65].

## Dopaminergic control of neuropeptide signaling

Both activation and inhibition of specific classes of neuropeptidergic cells by optogenetic activation of THD' cells suggests a dual role for dopamine possibly due to the presence of different classes of DA receptors. The *Drosophila* genome encodes four DA receptors referred to as Dop1R1, Dop1R2, DD2R and a non-canonical DopEcR [66]. Dop1R1, Dop1R2 and DopEcR activate adenylate cyclase and stimulate cAMP signaling whereas DD2R is inhibitory [66]. Cell specific differences among dopamine receptors have been observed in adults. Down regulation of *Dop1R1* on AstA and NPF cells shifted preference towards sweet food whereas down regulation of *DopEcR* in DH44 cells shifted preference towards bitter food [67]. In third instar larvae a dopaminergic-NPF circuit, arising from central dopaminergic DL2 neurons, two cells of which are marked by *THD'GAL4* (**Fig 2E**), motivates feeding in presence of appetitive odours [6,45]. The dopamine-neuropeptide axis identified here demonstrates a broader role for dopamine in regulating neuropeptide release and/or synthesis, in the context of larval feeding behaviour, perhaps similar to the mammalian circuit described above.

Of specific interest is the untimely upregulation of *ilp3* transcripts in *STIM$^{KO}$* larvae. Rescue of lethality in *STIM$^{KO}$* larvae either by bringing back activity to THD' neurons or by reducing ilp3 levels suggests an interdependence of Dopamine-Insulin signaling that is likely conserved across organisms [68–71]. Our data suggest that ilp3 expression is suppressed during the feeding and growth stages of larvae (**Fig 6H–6M**), and once enough nutrition accumulates expression of ilp3 is up-regulated, concurrent with a reduction in carbachol-induced Ca$^{2+}$ signals in THD' neurons, possibly followed by upregulation and release of ilp3. The idea of ilp3 as a metabolic signal whose expression is antagonistic to larval growth is supported by the observation that knock-down of ilp3 in the MNSc leads to larger pupae in wild type animals and larger larvae in *STIM$^{KO}$* (**Fig 6F and 6G**). To our knowledge this is the first report of ilp3 as a larval signal that is antagonistic to growth. Given that *Drosophila* encode a single Insulin receptor for ilp2, ilp3 and ilp5 [72] the cellular mechanism of ilp3 action remains to be elucidated. Possibly, ilps with different affinity for the insulin receptor stimulate different cellular subsets and/or different intracellular signaling mechanisms, including ecdysone signaling that is essential for

larval transition to pupae [72]. Interestingly, in $STIM^{KO}$ larval brains there is a significant increase in expression of the Insulin Receptor (**S1 Table**). Further studies are needed to fully understand ilp3 function in larvae.

Expression of other neuropeptides did not show significant changes in $STIM^{KO}$ larval brains (**S1 Table**), suggesting that for neuropeptidergic cells in the LNC and SEZ, dopamine signals alter release properties rather than synthesis. However, we were unable to identify specific neuropeptides for cells in the LNC and the SEZ that responded upon activation of THD'.

The importance of dopamine for multiple aspects of feeding behaviour is well documented in juvenile and adult mice [63,64]. Of interest are more recent findings linking dysregulation of dopamine-insulin signaling with the regulation of energy metabolism and the induction of binge eating [73,74]. The identification of a simple neuronal circuit where dopamine-insulin signaling regulates feeding and growth could serve as a useful model for investigating new therapeutic strategies targeted towards the treatment of psychological disorders for obesity and metabolic syndrome [73,75].

## Material and methods

### Fly rearing and stocks

*Drosophila* strains were reared on standard cornflour agar media consisting of 80 g corn flour, 20 g glucose, 40 g sugar, 15 g yeast extract, 4 ml propionic acid, 5 ml *p*-hydroxybenzoic acid methyl ester in ethanol, 5 ml ortho butyric acid in 1l at 25˚C, unless otherwise specified, under a 12:12 hr light: dark cycle. In all studies the *Canton S* (*CS*) strain was used as a wild-type control and CRISPR-Cas9 generated deficiency for *STIM* referred to as $STIM^{KO}$ served as a null mutant for the *Drosophila STIM* gene [22]. Details of other fly lines used are provided in Table 1 below.

### Staging

Synchronized larvae of the appropriate ages as described below were collected and transferred to agar less media containing yeast (4gm), sucrose (8gm), cornflour (16gm), Propionic acid (1ml) and 0.05gm of Benzoic acid in 1ml of absolute alcohol. The number of viable organisms and the developmental stage were scored at specific time points as mentioned below and in the figures and figure legends.

Larval staging experiments were performed to obtain lethality and developmental profiles of the indicated genotypes as described previously [76]. Depending on the experiment, timed and synchronized egg-laying was done either for 6h to allow development profiling at 60-66h, 80-86h, 128-136h, 176-182h and 320-326h after egg laying or for 2h at 35-37h(36h), 41-43h (42h), 47-49h(48h), 53-55h(54h), 59-61h(60h), 65-67h(66h), 71-73h(72h), 83-85h(84h), and 89-91h(90h) after egg laying for identifying a lethality window between 36-90h. Larvae were collected at either 60–66h or 35-37h after egg laying (AEL) in batches of 25 (for developmental profile) or 10 (for lethality window). They were screened and staged subsequently. Heterozygous larvae were identified using dominant markers (*FM7iGFP*, *TM6Tb*, and *CyOGFP*) and removed. Each batch of larvae was placed in a separate vial and minimally three vials containing agar-less media were tested for every genotype at each time point. The larvae were screened at the indicated time points for the number of survivors and stage of development, determined by the morphology of the anterior spiracles and mouth hooks [77]. Experiments to determine the viability of experimental genotypes and their corresponding genetic controls were performed simultaneously in all cases. Larval images were taken on the MVX10 Olympus stereo microscope using an Olympus DP71 camera.

**Table 1. Fly strains.**

| Fly line | Description | Source |
|---|---|---|
| Canton S | Wild type | |
| STIM$^{KO}$ | Null mutant for STIM gene generated with help of CRISPR-Cas9 gene editing technique | Generated in the lab |
| THGAL4 | Dopaminergic GAL4 driver marks TH (Tyrosine Hydroxylase) positive cells in brain and hypoderma | Serge Birman CNRS, ESPCI Paris Tech, France |
| THGAL80 | Inhibits Gal4 expression in Dopaminergic cells | Toshihiro Kitamoto, University of Iowa, Carver College of Medicine. |
| THD'GAL4 | Marks subgroup of Dopaminergic cells in larval brain | Mark N Wu, Johns Hopkins University, Baltimore |
| THC'GAL4 | Marks dopaminergic cells of larval brain | |
| C929GAL4 | Marks neuropeptidergic cells | P. H. Taghert, Washington University |
| c929LexA (DimmLexA) | Marks neuropeptidergic cells | Michael Texada, Janelia Farms, USA |
| UASSTIM+ | Stim wildtype cDNA under UAS control | Generated in the lab [81] |
| UASGRAB$_{DA}$ | A Genetically Encoded Fluorescent Sensor for Dopamine | Yulong Li, Peking University School of Life Sciences, Beijing, China |
| LexAopmCherry | Expresses mCherry under LexAop control | Claude Desplan, New York University, USA |
| UASmCD8GFP | Expresses membrane tagged GFP under UAS control | RRID: BDSC_5130 |
| UASKir2.1 | Prevents membrane depolarization | RRID: BDSC_6595 |
| UASGCaMP6m | Ca$^{2+}$ Sensor with intermediate kinetics expresses under UAS | RRID: BDSC_42748 from Douglas Kim |
| LexAopGCaMP6f | Ca$^{2+}$ Sensor with faster kinetics expresses under LexAop | RRID: BDSC_44277 |
| UASSyt-eGFP,Denmark | Syt-eGFP marks presynaptic terminals and Denmark (mCherry) marks post synaptic regions. | RRID: BDSC_33065 |
| UASCsChrimson | Optogenetically encoded cation channel that is activated at 625nm | RRID: BDSC_55136 |
| InscGAL4 | Marks neuroblasts of larval brain with GAL4 | RRID: BDSC_8751 |
| UASFUCCI | Marks different phases of cell cycle with fluorescent markers | RRID: BDSC_55100 |
| UASNaChBac | Increases sodium conductance and therefore activates the neuron | RRID: BDSC_9468 |
| UASERGCaMP-210 | ER specific Ca$^{2+}$ sensor | Cahir O'Kane, Cambridge University, UK |
| dsSTIM (III) | UAS-RNAi against Stim gene | VDRC_47073 |
| dsTH (III) | UAS-RNAi against Tyrosine hydroxylase (TH) gene | RRID: BDSC_25796 |
| UASDicer2(X) | Enhancer for RNAi | RRID: BDSC_24648 |
| MNScGAL4 (Dilp2GAL4) | Marks ilp producing MNSc cells of larval brain | RRID: BDSC_37516 |
| UASilp3$^{+}$ | Over expression of wildtype ilp3 gene | Ernst Hafen's lab |
| UASdsilp3 | RNAi for ilp3 gene | RRID: BDSC_33681 |
| UASDop1R1RNAi | RNAi for Dop1R1 receptor | RRID: BDSC_62193 |
| UASDopEcR RNAi | RNAi for DopEcR receptor | RRID: BDSC_31981 |
| UASDop2R2RNAi | RNAi for Dop2R receptor | RRID: BDSC_36824 |
| THD'GAL4, THGAL80/CyO | Recombinant line made for this study | |
| THD'GAL4, UASGCaMP6m/CyO | | |
| c929LexA, LexAopGCaMP6f, LexAopmCherry/TM6Tb | | |
| THD'GAL4, UASCsChrimson/CyO | | |
| STIM$^{KO}$; +; THGAL4/TM3SerG | Strains made for this study | |
| STIM$^{KO}$; THD'GAL4/CyOG; + | | |
| STIM$^{KO}$; THC'GAL4/CyOG; + | | |
| STIM$^{KO}$; UASSTIM/CyOG; + | | |
| STIM$^{KO}$; UASGCaMP6m/CyOG; + | | |
| STIM$^{KO}$; UASmCD8GFP/CyOG; + | | |
| STIM$^{KO}$; UASdsilp3/CyOG; + | | |
| STIM$^{KO}$; UASMNScGAL4/CyOG; + | | |
| UAS-STIM+/CyOG; UASGRAB$_{DA}$/Tb | | |
| UAS-STIM+/CyOG; UASERGCaMP-210/Tb | | |

## Feeding assay

Feeding assay was performed at specific developmental time points in larvae (40-44h, 58-62h, 80-84h AEL) of the specified genotypes. Larvae were placed in a 35mm punched dish with coverslip at the base thus creating a small depression in the centre of the coverslip. In this depression a coin sized cotton swab was placed containing 4.5.% of yeast solution with 0.25% eriogluasin dye (blue dye). For scoring the number of larvae that fed, 30 larvae per plate were taken and incubated in the feeding plate for 4hrs at 25˚C. Larvae were removed from the paste, washed, collected and scored for presence of blue dye (Dye$^{+ve}$) and absence of blue dye within the gut (Dye$^{-ve}$).

For quantification of ingested blue dye, 12–15 larvae were incubated for 2h in yeast paste with the blue dye. Larvae were removed from the yeast paste and 10 larvae with blue dye in the gut were washed, and homogenized in 50μl of cold 1xPBS. The homogenate was spun at 5k for 2 minutes in a table top Eppendorf centrifuge. The supernatant (2μl) was taken for quantification of protein using the Thermo scientific Pierce Protein assay kit, Cat#23227. Optical density (OD) at 625nm as a measure of ingested blue dye was measured from 30μl of the homogenate. Due to variation in larval sizes between control and experimental samples, the OD was normalized to whole larval protein concentration (μg/μl). OD was obtained using the SkanIt Software 6.1.1 RE for Microplate Readers RE, ver. 6.1.1.7. Larval imaging and processing was performed on the Olympus MVX10 stereo microscope using FIJI software.

## Larval imaging and measurement

Staged larvae from specified genotypes were collected at specific development time points, anesthetized on ice for 1h and mounted with ice cold HL3 buffer. The mounted larvae were imaged immediately using an Olympus MVX10 stereo microscope. For measurement of larval length from mouth hook to tip of the posterior spiracle FIJI software was used. A minimum of 10–15 larvae were taken per genotype for length analysis.

## Quantification of larval mouth hook contractions

Mouth hook contractions were measured by placing 1–3 appropriately staged larvae in a drop of 2% yeast solution in a petridish. Videos were taken for 30 seconds on an Olympus MVX10 stereo microscope. For each genotype a minimum of 10 animals were imaged. Mouth hook contractions were counted manually from the visualised videos.

## Pupal volume measurement

Pupal volume was measured by obtaining the width and height of each pupal image from the Olympus MVX10 stereo microscope. A formula for obtaining the volume of a cylinder, ($\pi r^2 h$) was applied to calculate the volume [78].

## Adult fly weight measurement

For weight measurement of adult flies, 10 flies (5 females and 5 males) of the appropriate genotype were taken 6–10 hr post-eclosion and weighed after placing them in a small Eppendorf tube. Thereafter, the weight of the same empty tube was measured. Fly weights were calculated by subtracting the weight of the tube from the total weight of flies + tube. A minimum of five such measurements were performed for each genotype.

## Immunohistochemistry

Larval brains were dissected in ice-cold 1xPBS and fixed with 4% paraformaldehyde in 1xPBS on the shaker for 20mins at room temperature. Fixed brains were washed with PBTx (0.3% TritonX-100 in 1XPBS) 3–4 times at 10minutes intervals, blocked with 5% normal goat serum (NGS) in PBTx for 2hrs at room temperature, and incubated with primary antibodies diluted in 5%NGS+PBTx at the appropriate concentration as mentioned below, overnight at 4˚C. Antibody solution was removed and re-used upto three times. Brains were washed with PBTx 3 times at 10 minute intervals followed by incubation with secondary antibodies at the dilutions described below, for 2hrs at room temperature and three washes in PBTx of 10minute intervals each. Brains were mounted in 70% glycerol diluted in 1xPBS. Confocal images were acquired by using FV3000 LSM and the Fluoview imaging software.

Following primary antibodies were used: Chick anti-GFP (1:8000, Abcam Cat#13970 RRID: AB_300798), rabbit anti-RFP (1:500, Rockland Cat# 600-401-379; RRID:AB_2209751), mouse anti-Prospero (1:100, DSHB Cat# Prospero (MR1A), RRID:AB_528440), rat anti-Dead-pan (1:400, Abcam–ab195172; RRID:AB_2687586), mouse anti-TH (1:5, ImmunoStar Cat#22941; RRID: AB_572268), mouse anti-bruchpilot (NC82) (dilution1:50, monoclonal, gift from Eric Buchner, University of Wuerzburg, Germany).

Following secondary antibodies were used: Goat anti-Chicken IgY (H+L), Alexa Fluor 488 (Thermo Fischer Scientific, Cat#A-11039; RRID: AB_2534096), goat anti-Rabbit IgG (H+L), Alexa Fluor 568 (Thermo Fischer Scientific Cat# A-11011; RRID: AB_143157), goat anti-mouse IgG (H+L) Alexa Fluor 633 (Thermo Fischer Scientific Cat#A-21052; RRID: AB_2535719), goat anti-rabbit IgG (H+L) Alexa Fluor 594 (Thermo Fischer Scientific Cat#A-11037; RRID: AB_2534095).

## Ex-vivo imaging of the larval brain

GCaMP signals were obtained from appropriately aged larval brains dissected from the specified genotypes and dissected in hemolymph like saline (HL3) (70mM NaCl, 5mM KCl, 20mM $MgCl_2$, 10mM $NaHCO_3$, 5mM trehalose, 115mM sucrose, 5 mM HEPES, 1.5mM $Ca^{2+}$, pH 7.2). Dissected brains were transferred to a 35mm punched dish with a cover slip adhered to the bottom. Brains were embedded in $\sim$5μl of 0.8% low melt agarose (Invitrogen, Cat#16520–100) and bathed in 86μl of HL3. Images were acquired as a time series on an XY plane at an interval of 2sec using a 20X-oil objective on an Olympus FV3000 inverted confocal microscope (Olympus Corp., Japan). For KCl stimulation, at the 40th frame, 7μl of HL3 was added and at the 80th frame 7μl of 1M KCl was added. The final concentration of KCl in the solution surrounding the brain was 70mM. For stimulation with Carbachol (Sigma Aldrich Cat# C4382), 10μl of HL3 was added at the 40th frame followed by 10μl of 100mM Carbachol at the 80th frame. Final carbachol concentration was maintained at 0.5mM $Ca^{2+}$ responses were imaged till the 300th frame (600sec).

Changes in ER-$Ca^{2+}$ were measured using an ER-GCaMP-210 strain [44]. The brain sample was prepared as above. Images were acquired as a time series on an XY plane at an interval of 1 sec using a 20X oil objective on an Olympus FV3000 inverted confocal microscope (Olympus Corp., Japan). For Carbachol stimulation, 10μl of HL3 was added at the 50th frame and 10μl of 100mM of Carbachol was added at the 100th frame. Final carbachol concentration was maintained at 0.5mM. Images were obtained for 600 frames (600 secs).

Larval brain expressing *MNSc>GCaMP* is used for Dopamine (DA) (Sigma, Cat#H8502) stimulation. Dissected brains were transferred to a 35mm punched dish with a cover slip adhered to the bottom. Brains were embedded in $\sim$5μl of 0.8% low melt agarose (Invitrogen, Cat#16520–100) and bathed in 80μl of HL3 having 1uM of TTX. Images were acquired as a

time series on an XY plane at an interval of 1.5sec using a 20X-oil objective on an Olympus FV3000 inverted confocal microscope (Olympus Corp., Japan). At the 30[th] frame, 10µl of HL3 was added and at the 60[th] frame 10µl of 1mM DA was added. $Ca^{2+}$ responses were imaged till the 250[th] frame (450sec).

## Analysis of Optogenetic signals

Larvae from the specified genotypes were reared in fly media containing 0.2mM ATR (All-trans retinal Sigma-Aldrich Cat#R2500). Brain samples were prepared as mentioned above. Images were acquired as a time series on an XY plane at an interval of 2 sec/frame using a 20X oil objective on an Olympus FV3000 inverted confocal microscope (Olympus Corp., Japan). For optogenetic stimulation of *CsChrimson*, a 633nm LED (from Thor labs) was used and GCaMP6f fluorescent images were obtained simultaneously using a 488nm laser line so as to measure changes in cytosolic $Ca^{2+}$ upon CsChrimson activation. Image acquired till 200[th] frames (400 secs).

A minimum of 6 independent brain preparations were used for all live imaging experiments and the exact number of cells imaged are indicated in the figures. Raw fluorescence data were extracted from the marked ROIs using a time series analyser plugin in Fiji. ΔF/F was calculated using the following formula for each time point (t): $\Delta F/F = (F_t/F_0)/F_0$, where $F_0$ is the average basal fluorescence obtained from the first 40 frames.

## RNA isolation and library preparation

Larval brains (15–20 per sample) were dissected from animals of appropriate genotypes and age (*CS*, 58-62h AEL; *STIM^KO*, 72-76h AEL), in ice cold phosphate buffered saline (PBS). Larval brain samples were transferred to tubes containing 300µl TRIzol (Invitrogen-15596018), and vortexed immediately for 10–15 secs. The vortexed samples were stored at −80˚C for further processing for up to one week. RNA isolation was done using Trizol, following manufacturer's protocol. RNA was run on a Bioanalyzer (Agilent) to ensure integrity. For each sample, 10ng of isolated RNA was used for cDNA synthesis using the SMART-Seq v4 Ultra Low Input RNA Kit, following manufacturer's protocol. The kit employs polyA tail complementary primer, template switching and extension by reverse transcriptase. Qubit dsDNA HS kit, following manufacturer's protocol, was used for assessing DNA concentration using 1µL of the cDNA sample. Further, Nextera XT DNA library kit (Illumina- FC-131-1024) was used for library preparation with 1ng of cDNA, following manufacturer's protocol. cDNA libraries were made from four independently isolated sets of brain RNA from each genotype. Libraries were pooled (2nM) at equimolar quantities and subjected to high depth sequencing in Illumina HiSeq 2500 (1 x 50bp).

## RNASeq analysis

FastQC and trimmomatic were used for QC of raw reads and adapter removal (if found) respectively. Raw reads were then mapped to *Drosophila* genome dm6 assembly using hisat2 [79]. The output BAM files were sorted and indexed using Samtools. The BAM files were used as input for htseq-counts. The htseq counts were then used as an input in DESeq2 (Bioconductor—R package) to obtain differentially expressed genes using default thresholds. Ggplot (R package) tool on R (version 4.1.2) was used to create the volcano plot in **S5A Fig**. Complete data for RNAseq is available at https://www.ncbi.nlm.nih.gov/geo/query/acc.cgi?acc= GSE202098.

## RNA isolation and quantitative PCR

RNA from 10–15 larval brains was isolated as described above for cDNA library preparation. RNA (1μg) was taken for cDNA synthesis. DNAse treatment and first strand synthesis was performed as previously described [80]. Reverse Transcription followed by PCR (RT-PCR) was performed in a reaction mixture of 25 μl with 1 μl of the cDNA. Quantitative real time PCRs (qPCRs) were performed in a total volume of 10μl with Kapa SYBR Fast qPCR kit (KAPA Biosystems) on a QuantStudio 3 Real-Time PCR system. Duplicates were performed for each qPCR reaction. Minimum of three biological replicates were taken for each qPCR reaction. *rp49* was used as the internal control. The fold change of gene expression in any experimental condition relative to wild-type was calculated as $2^{-\Delta\Delta Ct}$, where $\Delta\Delta Ct$ = (Ct (target gene) – Ct (rp49))$_{Expt.}$ – (Ct (target gene) – Ct (rp49))$_{Control.}$

Sequences of PCR primers used are as follows:

rp49
  F_5'CGGATCGATATGCTAAGCTGT3'
  R_5'GCGCTTGTTCGATCCGTA3'
dilp2
  F_5'CCATGAGCAAGCCTTTGTCC3'
  R_5'TTCACTGCAGAGCGTTCCTTG3'
*dilp3* F_5'ACTCGACGTCTTCGGGATG3'
  R_5'CGAGGTTTACGTTCTCGGCT3'
*dilp5* F_5'ACTCACTGTCGAGCATTCGG3'
  R_5'GAGTCGCAGTATGCCCTCAA3'

## Quantification and statistical analysis

All bar graphs and line plots show the means and standard error of means. In boxplots, horizontal lines in the box indicate median, box limits are from 25$^{th}$-75$^{th}$ percentiles, and individual data points are represented by closed circles (unless otherwise specified in the figure legends). Unpaired student t-Test (for two genotypes) and one way ANOVA followed by post-hoc Tukey's significance test (for data with multiple genotypes) was performed to calculate P values, given for all figures in **S2 Table**. All graphs were plotted using Origin 8.0 software. Origin 7.5 MicroCal, Origin Lab, Northampton, MA, USA N/A, Fiji Open access (RRID: SCR_002285). Diagrammatic representative images are made with help of Biorender website (https://app.biorender.com).

## Supporting information

**S1 Fig.** **(A-B)**. Number of viable organisms (mean ± SEM) at the indicated developmental stage of *CS* and *STIM*$^{KO}$ after egg laying. Number of sets (N) = 3, number of organisms per set (n) = 25. **(C)**. Confocal images of larval brains at the indicated time points expressing the FUCCI marker. Control genotype is *Insc>FUCCI* (top row) and the mutant genotype is *STIM*$^{KO}$; *Insc>FUCCI* (bottom row). Here late mitosis/G1 phase, S-phase and G2/early mitosis are marked by green, red, and yellow fluorescent indicators respectively. Scale bar = 20mm; n = 5 larval brains. **(D)**. Representative image of dye-fed larvae from *CS* and *STIM*$^{KO}$ at 80-84h AEL. Scale bar: 2mm. **(E)**. Bar graph showing the average number of Dye$^{+ve}$ (presence of blue dye in the gut) and Dye$^{-ve}$ (absence of blue dye in the gut) *CS* and *STIM*$^{KO}$ larvae at the indicated ages (mean ± SEM). Number of feeding plates per time point (N) = 3, number of larvae per plate (n) = 30. *$P < 0.05$, Student's *t*-test with unequal variances. P values are given in **S2 Table**. (TIF)

**S2 Fig. (A).** Stack bar graph showing the number of viable organisms (mean ± SEM) and their developmental stage at the specified hours after egg laying for the indicated genotypes. Coloured arrows mark bars that exhibit rescue of $STIM^{KO}$ upon expression of STIM$^+$ driven by *THGAL4* (dark green), *THC'GAL4* (blue), *THD'GAL4* (green) and restricted rescue in presence *of THGAL80* (red) with all appropriate genetic controls as indicated. Number of sets (N) = 3, number of organisms per set (n) = 25. **(B).** Stack bar graph showing the number of viable organisms (mean ± SEM) and their developmental stage at specified hours after egg laying for the indicated genotypes; *THD'/+*, *dicer;+;dsSTIM/+* (controls) and *THD'>dicer,dsSTIM*. Number of sets (N) = 3, number of organisms per set (n) = 25. **(C).** Quantification of weight of 10 flies from indicated genotypes. Box plots show the 25$^{th}$ and 75$^{th}$ percentiles with median (bar), mean (square) and each circle represents one set. Each set consists 10 flies of which 5 are females and 5 are male adult flies collected at 6-8h after eclosion. Number of set (N) ≥5. Alphabets indicate different statistical groups. In all panels significant changes between relevant genotypes at the indicated stages were calculated by one way ANOVA followed by post-hoc Tukey's test. P values are given in **S2 Table**.
(TIF)

**S3 Fig. (A).** Representative confocal images of larval brains showing THD' neurons marked with anti-GFP (green) and anti-TH (red) from control (*THD'>mCD8GFP*), $STIM^{KO}$ ($STIM^{KO}$; *THD'>mCD8GFP)* and rescue ($STIM^{KO}$;*THD'>mCD8GFP*, STIM$^+$) animals at the indicated developmental time points. Scale bars = 20μm. **(B).** Numbers of TH positive cells (magenta) and THD' cells (green) in the larval CNS of the indicated genotypes. Cells were quantified from *THD'>mCD8GFP* (control) at 58-62h and 80-84h and from $STIM^{KO}$;*THD'>mCD8GFP* ($STIM^{KO}$) and $STIM^{KO}$;*THD'>mCD8GFP*, STIM$^+$ (rescue) at 80-84h. The numbers of TH$^{+ve}$ and GFP$^{+ve}$ cells were counted manually and were no different among four hemi-lobes from four brains of a single genotype and among all brain hemi-lobes of all genotypes. Hence the absence of error bars. **(C).** Stack bar graph showing the number of viable organisms (mean ± SEM) and their developmental stage at specified hours after egg laying for the indicated genotypes *THD'/+*, *dsTH/+* (controls), and *THD'>dsTH*. Number of sets (N) = 3, number of organisms per set (n) = 25. Significant changes were calculated by one way ANOVA followed by post-hoc Tukey's test. P values are given in **S2 Table**. **(D).** Representative images of the central brain (left panels) indicating the region of focus (boxed), followed by images of THD' cells with Ca$^{2+}$ transients at the indicated time points after addition of a depolarizing agent (KCl, 70mM). Ca$^{2+}$ transients were measured in the indicated genotypes: *THD'>GCaMP6m* and *THD'>dicer;dsSTIM,GCaMP6m* by measuring changes in the intensity of GCaMP6m fluorescence. Scale bar = 20μm. **(E).** Quantified changes in GCaMP6m fluorescence (mean ± SEM of ΔF/F) from THD' neurons of the indicated genotypes. Number of brains, (N) ≥ 6, number of cells, (n) ≥11. **(F).** Peak intensities of GCaMP6m fluorescence (ΔF) in THD' cells from the indicated genotypes. Box plots show 25$^{th}$ and 75$^{th}$ percentiles, the median (bar), and mean (square) of ΔF of each cell (small circles).
(TIF)

**S4 Fig. (A-B).** Altered excitability in THD' neurons by expression of either a mammalian inward rectifying potassium channel (Kir2.1) or a bacterial sodium channel (NachBac) affects larval developmental progression and viability. Stack bar graphs show the number of viable organisms (mean ± SEM) and their developmental stage at the indicated hours after egg laying (top) for the indicated genotypes. Number of sets (N) = 3, number of organisms per set (n) = 25. P values calculated after one way ANOVA followed by post-hoc Tukey's test are given for relevant stages and genotypes in **S2 Table**. **(C).** Representative images of the central brain (left panels) indicating the region of focus (boxed), followed by images of THD' cells with Ca$^{2+}$

transients at the indicated time points after addition of mAChR agonist (CCh 100μM). $Ca^{2+}$ transients were measured in the indicated genotype by measuring the changes in the intensity of GCaMP6m fluorescence. Scale bar = 20μm. **(D)**. Quantified changes in GCaMP6m fluorescence (mean ± SEM of ΔF/F) from THD' neurons of the indicated genotypes. Number of brains, (N) ≥ 4, number of cells, (n) ≥12.
(TIF)

**S5 Fig. (A)**. Upregulated (magenta colour, log2fold ≥ +1; p<0.05) and downregulated (green, log2fold ≤ -1; p<0.05) gene sets in $STIM^{KO}$ larval brains (72-76h AEL) as compared to *CS* larval brains (58-62h AEL), depicted as a volcano plot. N = 4. Also see **S3 Table** for gene names and their mean expression levels. **(B)**. Larval developmental profile with overexpression of ilp3 in MNSc (red arrow) appears similar to knockdown of ilp5 (green arrow). Stack bar graph showing the number of viable organisms and their developmental stage at the specified hours after egg laying for the indicated genotypes. N = 3 sets with 25 larvae in each set. **(C)**. Knockdown and overexpression of ilp3 in the MNSc affects adult weights in opposing directions. Quantification of weight of 10 flies from the indicated genotypes. Box plot shows the 25[th] and 75[th] percentiles with median (bar), mean (square). Each circle represents one set consisting of 10 flies in each of which 5 females and 5 males were collected at 6-8h post- eclosion. A minimum of 5 sets were quantified for each genotype. Alphabets indicate statistically different groups based on P values calculated after one way ANOVA followed by post-hoc Tukey's test, given for relevant stages and genotypes in **S2 Table**.
(TIF)

**S1 Video. Mouth hook contractions in *CS* larvae (70-74h).**
(AVI)

**S2 Video. Mouth hook contractions in $STIM^{KO}$ larvae (70-74h).**
(AVI)

**S3 Video. Mouth hook contractions in *CS* larvae (82-84h).**
(AVI)

**S4 Video. Mouth hook contractions in $STIM^{KO}$ larvae (82-84h).**
(AVI)

**S5 Video. Mouth hook contractions in *THD'>dsTH* larvae (82-86h).**
(AVI)

**S6 Video. Mouth hook contractions in *THD'/+* (control) larvae (82-86h).**
(AVI)

**S7 Video. Mouth hook contractions in *dsTH/+* (control) larvae (82-86h).**
(AVI)

**S1 Table. Expression status of genes encoding neuropeptides in $STIM^{KO}$ brains.**
(DOCX)

**S2 Table. P values for main and supplementary figures.**
(DOCX)

**S3 Table. Contains the list of differentially expressed genes, identified from the RNAseq experiment performed with CS and $STIM^{KO}$ larval brains (related to Figs 6 and S5) as an Microsoft Excel sheet.**
(XLSX)

**S4 Table. Contains the source data for all figures and supplementary figures as Microsoft Excel sheets.**
(XLSX)

## Acknowledgments

We thank Fly base for access to various databases, Bloomington and VDRC fly stock centres, Fly Facility at NCBS for fly lines,NCBS Central Imaging and Flow Facility (CIFF) for confocal imaging, NCBS Next Genome Sequencing (NGS) facility for help with RNAseq and NCBS lab support for timely help.

## Author Contributions

**Conceptualization:** Nandashree Kasturacharya, Gaiti Hasan.

**Formal analysis:** Nandashree Kasturacharya, Jasmine Kaur Dhall, Gaiti Hasan.

**Funding acquisition:** Gaiti Hasan.

**Investigation:** Nandashree Kasturacharya, Jasmine Kaur Dhall.

**Project administration:** Gaiti Hasan.

**Resources:** Nandashree Kasturacharya, Gaiti Hasan.

**Supervision:** Gaiti Hasan.

**Visualization:** Jasmine Kaur Dhall.

**Writing – original draft:** Nandashree Kasturacharya, Jasmine Kaur Dhall.

**Writing – review & editing:** Gaiti Hasan.

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
