## [Decision Letter · Decision Letter 0]

3 Nov 2022

Dear Gaiti,

Thank you very much for submitting your Research Article entitled 'A STIM dependent dopamine-insulin axis maintains the larval drive to feed and grow in Drosophila' to PLOS Genetics.

The manuscript was fully evaluated at the editorial level and by independent peer reviewers. As you will see, the reviewers all agree that the phenotype is interesting, but have a number of serious concerns regarding mechanism/interpretation (reviewer 1), the link or lack thereof between STIM and ILP-positive cells (reviewer 2), and the lack of key experiments to show a cell type-specific rescue, as well as careful feeding assays (reviewer 3). All three reviewers also comment that the amount of work necessary to satisfactorily address the concerns is likely to go well beyond what might be expected in even a major revision.

The manuscript and the reviews have now been discussed among members of the editorial board. We agree with the concerns expressed by the reviewers and are willing to consider a revised manuscript, but caution that its potential success would depend on satisfactorily addressing the concerns. 

Based on the reviews, we will not be able to accept this version of the manuscript, but we would be willing to review a much-revised version with a resubmission time of 6 months. We cannot, of course, promise publication at that time.

If you decide to revise the manuscript for further consideration at PLOS Genetics, please aim to resubmit within the next 180 days, unless it will take extra time to address the concerns of the reviewers, in which case we would appreciate an expected resubmission date by email to plosgenetics@plos.org.

We are sorry that we cannot be more positive about your manuscript at this stage. Please do not hesitate to contact us if you have any concerns or questions.

Yours sincerely,

Gregory Barsh

Editor-in-Chief

PLOS Genetics

Gregory Copenhaver

Editor-in-Chief

PLOS Genetics

Reviewer's Responses to Questions

**Comments to the Authors:**

**Please note here if the review is uploaded as an attachment**.

Reviewer #1: Like other juvenile organisms, food intake, the quality and quantity of food ingested, can influence Drosophila melanogaster juvenile size. This study describes a subset of dopaminergic neurons marked by a THD-Gal4 that appears to maintain the larval motivation to feed. Dopamine release from these neurons requires the ER Ca2+ sensor STIM. Early larvae with loss of STIM stop feeding, grow and eventually die. The authors show that the expression of STIM in the THD neurons rescues the feeding, growth and viability of STIM null mutants. STIM maintains excitability and can partially rescue the dopamine release from THD neurons. The authors used optogenetic stimulation of THD neurons to identify connectivity to neuropeptidergic cells and found that loss of STIM in THD cells alters the developmental profile of specific insulin-like peptides, downregulating Ilp2 and Ilp5, and upregulating Ilp3. Silencing Ilp3 partially rescues STIM null mutants, and strikingly overexpression of ilp3 in larvae affects development and negatively growth. Together these studies identify a STIM-dependent dopamine-ILP circuit that regulates developmental changes in larval feeding behaviour.

The data are very interesting and intriguing; the figures are well-constructed, solid, and well-presented. Unfortunately, there are important problems to be clarified about the mechanism, which is still obscure. There is a lack of experiments that address the problem directly. The overall data suggest that development and growth are arrested first, and then feeding is finalised. The authors suggest that feeding is the main phenotype, but they do not show that STIM or Dopamine has an instructive role, and they do not show clear data that integrate feeding with growth in a direct way. The poor rescue of dsIlp3 raises important questions, particularly given the strong genetic background of transgenes. The rescue of NaChBac or the role of Dopamine release are also poor or unclear. The introduction needs some work and focus on the juvenile stage. Key publications are missing, and others that are not relevant are discussed at length.

Major comments:

Given the great differences in the control of feeding behaviour between larvae and adult flies, the citations of articles and circuits, sometimes related to adults and sometimes to larvae, and sometimes without clarifying if they refer to larvae or adults, are confusing, giving rise to possible misinterpretations. Authors need to rewrite and restructure their introduction and also avoid over-interpretations of published articles. Next, I comment on cited articles in the introduction, which need to be fixed or toned down.

Line 49-50. It seems reasonable that the appetite is regulated generally depending on the nutritional status. However, these two articles they cite to support the claim that increased appetite is associated with accelerated growth are articles that show genetic manipulations or sensing of the internal state occurring "despite normal feeding behaviour" (PMID: 14505573). These nutrient-sensing pathways inhibit larval feeding, further showing that appetite, sensing internal states, and growth can be genetically separate. Along the same lines, the work of Layalle and Leopold (PMID: 18854141) deals with the intrinsic mechanisms that control larval growth and puparium time.

Wu et al. show that NPF neural system is not an essential part of the basal feeding machinery but is crucial for normal food response under deprived circumstances (motivational feeding). Silencing by RNAi "did not decrease the size and weight compared with wild-type flies". Therefore the statement in the current manuscript that "Mutants for short Neuropeptide F (sNPF) affect body size by regulating food intake [14]" seems incorrect.

Line 54-55: "Allatostatin A (Asta), Drosulfakinin DSK) and Tackykinin (TK) access the internal metabolic state and terminate feeding based on the energy state of the organism [8–11]". These circuits relate to adult flies; I think they do not control larval growth. Martelli et al. are also regulation of feeding in adult flies. SIFamide is adult feeding behaviour.

Reference 15, Schoofs A, Hückesfeld S, Pankratz MJ relates to ex vivo studies of feeding motor neurons.

Reference 16, Youn H, Kirkhart C, et al. relates to octopaminergic neurons that promote feeding initiation in adult Drosophila melanogaster.

References 22 and 23, refer to modulations but not to stop feeding.

The idea that what stops feeding is the animal's nutritional status is interesting, but not supported by published data (yet). The current view is that larval size determines the time of feeding cessation and food exit (Stern and Emlen, 1999) and is mechanistically related to an ecdysone pulse.

Therefore, the statement: "Multiple neuronal circuits thus integrate sensory inputs and internal metabolic state to maintain feeding and appropriate feeding behaviour in larvae and adults" should be toned down.

The authors should avoid over and misinterpretations of papers, perhaps eliminating the adult papers or clearly defining when they refer to feeding regulation during growth or in adult animals.

Figure 1H. The size and feeding rate of STIM-KO mutants are similar to early 60-72 h AEL larvae that their chronological age, raising the question whether the reduced feeding is because the mutant are arrested at a low feeding stage.

Please quantify the weight or size of the larvae over time. Although it said the images are representative, the images in Figure 1C and 1H show larvae of very different sizes, with the mutant larvae in H much larger than those in C.

Figure 1I and IJ and Figure 1C suggest than mutant larvae first stop growing and later stop feeding. Please comment. It might be informative to measure larvae after feeding to associated feeding behaviour and growth.

Figure 2 rescue.

STIM rescue in THC'GAL4-labeled neurons is described as poor rescue (5+1 adults; S2A Fig), but rescue by NaChBac or dsIlp3 is even poorer.

The rescue by STIM overexpression is absolute. Nevertheless, according to Figure 4, which measures Dopamine release, the dynamics of dopamine release do not reach the control level, which questions the importance of dopamine release. It also questions the physiological role of dopamine-insulin in the process, as inferred from the title.

Observing dopamine release, together with the NaChBac data, the THC data, and the poor dopamine receptor defect (only developmental delay), suggests that the underlying (main) defect may be another uncharacterized one. This problem needs to be sorted out.

Please correct Figure S1 labelling. Confocal (C) and (D) are representative images of dye-fed larvae from CS and STIMKO at 80-84h AEL.

According to the authors not all neurons in THD are dopaminergic. Therefore, it should be more correct to refer in Figure 5C. to THD cells rather to dopaminergic neurons, as the contribution of non-TH neurons in the optogenetic experiments is not controlled.

Figure S6. The number of adults rescued are n=3 in three experiments of n=10 or n=30? It seems rather low. Given the role of Ilp3 in regulating other Ilps and that Ilp3-KO has no phenotype, the authors should present more data on the possible physiological role of Ilp3 and if feeding is affected or only growth and how an Ilp silencing promotes growth.

Figure S1B shows viability, but a curve of changes in larval growth over time would be informative to define the timing of feeding and growth defect.

"The momentum of larval growth is maintained by constant cell division and cell growth. Larval growth occurs by cell growth, and until this moment in the article, the authors had not monitored the cell division of internal organs. Therefore, "Cessation of growth in STIMKO larvae (Fig 1D), suggested a deficit in cell division" seems strange. It would rather suggest a defect in developmental progression or in cell growth.

Line 127. First, the authors stated that STIM-KO larval neuroblasts existed quiescence as the control brains, but cell division slowed later. Therefore, there is no defect in nutritional input because the "brain sparing" process enables the brain to continue proliferation even in completely starved larvae (once quiescence is released). The impact of slow cell division on neuroblasts and systemic growth is not further discussed. The THD circuit appears to be unaffected, so cell division in these neurons is not affected by reduced feeding? It is, therefore, unclear what the authors attempt to communicate here.

"Based on these results it appears that STIMKO larvae gradually stop feeding from early 2nd instar larvae and the consequent nutritional deficits prevent normal growth."

STIM-KO stops growth or has reduced growth apparent in 60 h AEL, suggesting that slow growth or developmental arrest is an early defect, perhaps even earlier than feeding. The data presented do not clarify what the cause is and what the effect is.

Since the STIM-KO defect is more complex than what has been reported when starving wild larvae of that age, the authors should consider all possibilities and present observations without assuming or over-interpret that one defect is the cause of the another defect. Could feeding L-DOPA rescue the growth or feeding defect?

Lines 133-135 should be toned down or eliminated.

Related to THD’>dsSTIM animals. These animals exhibit delayed larval growth and reduced feeding (Fig 2I-2K; S2B Fig) but no larval lethality. It is relevant to present data on the adults, their size, and weight. What is the impact of delayed larval growth and reduced feeding?

Line 191: "A few cells (70%) responded…." I think that 70% cannot be described as "a few cells"

Line 199: The rescue with NaChBac supports the role of STIM-KO in the excitability of THD neurons. However, the explanation of variability due to a stochastic effect of NaChBac is an ad hoc interpretation and an inference of the incomplete effect. It could be that excitability is only part of the phenotype.

How does STIM control larval growth, feeding and Dopamine release?

Line 228 "Though ER-Ca2+ release, in response to CCh could be measured in just 7 out of 23 cells, the subsequent step of ER-store refilling, presumably after Store-operated Ca2+ entry into the cytosol through the STIM/Orai pathway, could be observed in all control THD' cells… and absence in THD' neurons from STIMKO brains. However, the release of Dopamine is reduced, not absent. The dynamics of STIM-KO, THD>STIM dopamine release suggests additional mechanisms not only defect in dopamine release. Please comment or clarify how to interpret the data in this figure.

Why have authors not tested the potential role of dopamine release more directly? The effects of dopamine receptors do not clarify whether the main effect of STIM-KO is related to excitability/dopamine release. Line 272. Knockdown of Dop1R1 in peptidergic cells suggests that Dopamine regulates development time, but this datum does not indicate whether there is a control/defect on lfeeding or arval growth deficits that would mimic STIM-KO.

The experiments with CCh stimulation are interesting but do not add or explain the link between feeding, feeding cessation, and larva growth and defects in STIM-KO animals.

TransTango in leaky in the larval brain: In the original paper of TransTAngo (https://doi.org/10.1016/j.neuron.2017.10.011) "the sensitivity of trans-Tango (in the larval brain) may lessen in circuits with sparser synaptic contacts. Please comment on this issue, as this is relevant to the validity of the presumed synaptic partners.

Lines 300-305. How can Ilp3 silencing rescue larval growth in animals with low levels of Ilp2 and Ilp5? These neuropeptides act through the same receptor. Could InR manipulation rescue STIM-K?

Does ILP3 overexpression in ILP2 neurons reproduce some of the defects of STIM-KO? How Dopamine release inhibits ILP2 and ILP5 and stimulates ILP3?

Lines 304-311: These conclusions are overinterpretations. Increased feeding is not the same as extended feeding. The authors must measure the feeding rate of Ilp3 in the STIM-KO animals.

"A small proportion of STIMKO larvae rescued by ilp3 knockdown appear significantly larger than control larvae (Fig 6F and 6G), suggesting excess feeding in the absence of ilp3." This suggestion must be verified with data. If not supported by data, this is a suggestion that Ilp3 may regulate stop feeding. Previous work has already shown that brain Ilps can inhibit feeding. However, given that Ilp3 is increased and Ilp2 and Ilp5 are decreased, it is important to assess the net effect. The authors must include the effects of silencing Ilp5, as a control of the effect. Ilp2 silencing is likely lethal.

"This idea is further supported by over-expression of ilp3 in MNSc [50] where we observed smaller sized larvae (Fig 6H and 6L), suggesting reduced feeding due to untimely over-expression of ilp3, and a delay in larval moults from 2nd to 3rd instar stage (Fig 6J).

Prolonged feeding can be a defect in development timing and ecdysone production or levels, which the authors have not tested.

Without testing the impact on feeding rate, the conclusions regarding Ilp3 are overinterpretations: "Taken together, these data show that loss of dopamine signals from THD' cells changes the normal expression profile of ilp3 in the MNCs and demonstrate that in normal development Ilp3 is one of the neuropeptides that signals an end to larval feeding prior to pupariation. Note that Ilp3-KO animals show normal larval growth.The discussion of the dopamine-insulin axis in rats is interesting, but is it related to the juvenile stage? Neither mice nor young rats stop feeding when they reach sexual maturity, so it is vague what the established parallelism refers to.The cessation of feeding before metamorphosis is a peculiarity of metamorphosis and is a genetic program. Perhaps the authors should test the STIM-dopamine-insulin axis in adult flies related to adult hunger/satiation.

Reviewer #2: The paper, entitled "A STIM dependent dopamine-insulin axis maintains the larval drive to feed and grow in

Drosophila" describes a new neuronal circuit that regulates larval feeding. Dopaminergic cells in the CNS are essential, which function via the Stromal Interaction Molecule (STIM) on neurosecretory cells that produce the insulin-like peptide ilp3. This circuit is responsible for the age-appropriate regulation of feeding behavior in the Drosophila larva.

Before the paper can be published, I have a few criticisms that the authors should address.

General concerns:

1) The anatomical description of CNS dopaminergic neurons has major gaps. The authors seem to have missed several important publications here. The paper by Selcho et al. 2009 (PlosOne) describes the DL1 and DL2 cluster neurons and the complete larval dopaminergic system in great detail at the single cell level. In addition, even connectome data are available. The DL1 cluster neurons are included in Eichler et al. 2017 (Nature) and Saumweber et al. 2018 (Nature Commun). Here, they are referred to as DAN-c1, DAN-d1, DAN-f1, and DAN-g1. The complete input and output circuits of DL1 dopaminergic neurons were also described at the synaptic level by Eschbach et al. 2021 (Nature Neurosci). A very extensive anatomical description results from this work, which unfortunately has not found its way in here so far. Unfortunately, these studies also strongly suggest that there is no direct synaptic connection between the DL1 cluster and neurosecretory ILP3 cells.

2)

In the above work and other extensive studies from the Gerber, Thum, and Zlatic laboratories, the dopaminergic DL1 neurons are shown to specifically innervate the larval mushroom body and not neurosecretory cells. They are responsible for punishment learning by mediating the negative teaching signal to the mushroom body during olfactory conditioning. This function is also found in adult flies and even in other insects. Important reviews and work on this topic can also be found from the Waddell, Preat, Davis, Rubin laboratories. The work would benefit greatly if these efforts were considered as well.

Specific concerns:

Figure 4A:

The description of the anatomy here is unfortunately incorrect. Based on the single cell description, we already know that the cells from the DL1 cluster innervate the mushroom body. Thus, the middle area shows the innervation of DL1 neurons to the larval mushroom body. The second area is very likely based on basolateral innervation of DL2 neurons. The third area is really incorrectly described here. This area is clearly within the CNS. It clearly overlaps with the NC82 staining in blue. It is an expression in the basomedial CNS that can originate from DL1 and/or DL2 neurons. In addition the labels are messed up and are wrongly described in the figure legend.

Figure 5 - Line 251-253:

I'm sorry, but I really disagree here. I see no evidence that the cells of the Transtango sample are identical to that of C929. First, the connectome information for the downstream partners of the DL1 cluster is now available. Based on the synaptic reconstruction (Eschbach et al. 2020) of the DL1 downstream circuit, no neuropeptidergic cells are identified. Second, single-cell analysis of DL2 cells also reveals no inervation of the pars intercerebralis and the neurosecretory cells loacated there. Third, the position of the transtango cells labeled with THD does not overlap with that of the cells labeled with C929. Fourth, when the size of the cells is carefully checked, it is also clear that they are different in size. Fifth, for such a statement, one really needs a co-labeling of the cells of THD-transTango and C929 in one animal. Separate evaluation of two brains from different animals is not acceptable for such a statement. Sixth, the analysis of receptor expression shows (Selcho et al. 2009) that there is no expression of Dop1R in the neurosecretory cells of the pars intercerebralis.

Based on these arguments, I unfortunately do not see a synaptic link between dopaminergic and ILP3 cells. Similarly, I lack the data to directly compare C929 Gal4 expression with ILP3 cells.

This will require further analysis showing a clear link. However, the data to date almost exclude such a link.

Therefore, I do not believe that the results are evidence for connectivity. In fact, it may not be necessary, since dopamine as a neuromodulator via volume transmission can act over longer distances.

In addition, there is a wonderful description of the anatomy, function, and synaptic connectivity of the larval pars intercerebralis - which you may see here in C929. It is composed of DMS, IPCs, and DH44-positive cells. This work was published by Huckesfeld et al. 2021. Similarly, there are now several publications to obtain specific lines also for neuropeptidergic cells (e.g. Deng et al. 2019, Neuron).

Figure 3A:

This is crucial - are you really sure that you have measured the signal from the same neurons? What if DL1 and DL2 neurons behave differently? There is a lot of data showing that DL1 neurons innervate the mushroom body and transmit reinforcement signals to the mushroom body. Especially in the case of aversive signals. The DL2 neurons, on the other hand, seem to be involved in food intake as you mention in the discussion. So what if the difference in signaling is due to differences between DL1 and DL2? The second row seems to show two DL2 neurons. The first row three DL1 neurons. But what is seen in the third row? Can they show more clearly which cells are involved here? Can't they show all 5 cells? Are there differences between the individual cells which allow to describe the cells individually? Is this maybe also true for the data shown in figure S3D?

Introduction LIne 51 - 68:

Would it be possible to restructure this part? It reads like a loose enumeration of results. However, it is somewhat confusing because it is rather unstructured and does not clearly distinguish between larval and adult data. It is often not clear whether the results apply to both or only to one developmental stage.

Figure 1:

The effect is clear - but why do the STIM KO images not focus on the thoracic ganglion? in particular, the last pciture shows both brain hemispheres and not the same part of the thoracic segment as the corresponding control. Also, the second column is displaced and partially encompasses the brain hemispheres.

The figure legend for Figure S1D is missing. The scale bar is also missing for the images.

What does Dye-ve and Dye+ve mean? I found the info in the material and method section, but it is missing in the figure legend.

Line 147 - 150:

I'm not sure if the number of viable adults is a good way for preseting the data. I guess you have to provide at least some ratio or the info that these are 20 out of 25 animals.

Line 151: Typo: mCD8-GFP

Line 153: Please cite the original data on the anatomy of the larval dopaminergic system (mentioned above).

Figure 2: Please use the labels DL1 and DL2 also in the picture of the brains. It would really help to keep the same descriptions in the text and figures.

Is there also some inter individual variability in the expression of THD-Gal4? It seems that there are only two DL1 neuron and no DL2 neuron on the other side of the brain.

LIne 175- 176:

How is this calculated? In the supplement it says exactly the opposite. Why are there no error bars, why are the results all exactly the same? How many brains were really used? Based on the images, I see a lower signal in the STIM KO, especially with two DL2 neurons and one DL1 neuron.

Reviewer #3: In animals, the ability to initiate, maintain, and terminate feeding when nutrients are available is essential for normal development, homeostasis, and fertility. Reduced feeding is associated with a smaller body size and lower fertility, and increased feeding is associated with obesity. Understanding the mechanisms behind the normal regulation of feeding behaviour is therefore an important biological question.

In this manuscript, the authors found that whole-body loss of STIM, an ER calcium sensor, led to a reduction in body size, a severe developmental delay, and the later death of larvae. In particular, the transition from the second to the third instar stage of larval development was impaired. The authors show that animals with whole-body loss of STIM had defects in thoracic neuroblast cell division, and that STIM mutants had reduced food intake compared with control larvae.

To determine the cell type in which STIM might act to mediate these effects the authors partially rescued viability in animals with a functional STIM protein in TH-GAL4-positive neurons, which are dopaminergic. In particular, the brain-specific THD’-GAL4 neurons were mostly responsible for this rescue of viability, and the authors observed improved food intake in these larvae. Importantly, loss of STIM in THD’ neurons delayed larval growth and reduced feeding behavior.

The authors then show that whole-body loss of STIM at later, but not earlier, larval stages caused defects in their THD’ neuron calcium response, which could be rescued by expressing STIM in those neurons. When the authors knocked down STIM in THD’ neurons they found that some animals had a complete failure of calcium responses while others had relatively normal responses, indicating heterogeneity in the neuronal response to STIM loss. The authors partially rescue viability of STIM knockout animals by overexpressing NaChBac in THD’ neurons, and show that STIM is important for dopamine release from the THD’ neurons.

The authors go on to show that the THD’ neurons are functionally connected to peptidergic neurons, with some peptidergic neurons activated by THD’ neuron activity and others inhibited. That dopamine was important for activation of peptidergic neurons was indicated by an experiment in which loss of dopamine receptors on C929-GAL4-positive neurons caused a developmental delay.

The authors show that whole-body loss of STIM caused a decrease in brain levels of ilp2, ilp5 and ilp3, with ilp3 upregulated by more than 5 fold. The authors were able to rescue these altered ilp mRNA levels by THD’-specific expression of STIM, and also partially rescues larval development. In animals with knockdown of ilp3 in the insulin-producing neurons the animals are bigger, and overexpression of ilp3 in these same cells reduced body size.

Based on this data, the authors propose a model in which STIM acts in THD neurons to promote calcium homeostasis and excitability of these neurons downstream of cholinergic inputs. This allows dopamine to be released which binds to Dop1R on the surface of peptidergic neurons, which stimulates (at least in the mNSCs) changes in ilp levels to alter body size.

The underlying idea behind much of this work is very elegant, and some of the findings are highly interesting and suggestive. However, there are multiple methodological inconsistencies, incorrect genotypes, statistical errors, and assumptions that will require a large amount of work to remedy ahead of publication.

Major issues

1. Much of the work describing a cell type-specific role for STIM relies upon rescuing STIM to the THD’ neurons in a whole-body knockout animal. As far as I can tell, in the majority (or all) the cases the control animal is Canton-S (no transgenes), the rescue genotype is STIMKO with THD’ expression of STIM, with STIMKO as the mutant. Unfortunately, the genotypes used by the authors do not support their claim of a cell type-specific role for STIM for the following reasons.

In Drosophila, to show a cell type-specific rescue, it is essential to have the following controls: mutant, GAL4 line in a mutant background, UAS line in a mutant background, UAS+GAL4 in a mutant background. With phenotypes like body size, feeding, and larval development, all of which are affected by a huge number of genes, it is essential to know that neither genetic background “rescues” the phenotype on its own, and that the “leaky” expression of the UAS (which occurs due its insertion site) does not rescue the phenotype either (because then it is not a cell type-specific rescue). Without these proper controls, the authors cannot convincingly show that STIM in the THD’ neurons affects the phenotypes they have shown in the paper. A revised version of the paper must replicate all phenotypes in the paper with the proper controls.

2. The main message that the authors lay out in the paper is about control of feeding behavior. There are two issues that prevent the authors from making accurate conclusions about feeding (see also next point). First, for many genotypes the authors do not actually measure this phenotype. Instead, the authors rely heavily on developmental delays and body size to make conclusions about feeding (THD’>Kir2.1;THD’>NaChBac;C929-GAL4>Dop1R1; MNSC ilp3 overexpression and knockdown; ilp3 knockdown in STIMKO). This is not appropriate; while feeding will certainly affect body size and larval development, many things that affect larval development and body size have no effect on feeding. In the revised manuscript these genotypes must have accompanying feeding behaviour (and mouth hook contraction data see below) to support a role in feeding.

3. Second, and the assays used to monitor feeding are not ideal. A much better and easier assay is to count mouth hook contractions across 15-20 animals for 30 sec per animal. This will tell you whether the animal is motivated and trying to feed independently of how much food it can hold in its gut. Another factor that may complicate the feeding phenotype is whether the larvae have initiated wandering prematurely. This can be easily monitored by asking where the animals are found (are they in the food still but not eating or are they leaving the food where there is none to consume?). These tests will help the authors make more accurate conclusions about the phenotype of the STIMKO larvae is caused by precocious wandering (as described in Britton et al 2002) or reduced feeding.

4. RNAseq data of STIMKO and CS brains must be presented for evaluation by reviewers and readers. There may be large trends that are visible when an unbiased analysis of the data is presented. Unfortunately this was not made available.

5. There must be connections made between the dopamine receptor in MNSC and the insulin responses in STIMKO animals to fully support the authors’ model that dopamine from THD’ neurons mediates the changes in ilp levels in MNSC. The authors must determine whether phenotypes caused by loss of Dop1R1 in THD’ neurons (including feeding and developmental) are rescued by overexpression/knockdown of ilp levels.

6. Statistical tests must be adjusted throughout – according to the methods the authors used t-tests. In most cases (any time there are more than two genotypes, which is most experiments, the essential test is a one-way ANOVA using multiple comparisons to obtain p-values. All of the comparisons should be shown on the graphs (not just one comparison that is significant, some comparisons might be not significant or ‘ns’). The authors must determine whether the rescued genotype is significantly different from all control genotypes (see note 1 on correct genotypes).

Minor problems

1. The reason for the heterogenenous response of THD’>dsSTIM animals’ calcium response to KCl is unclear. Because there are a number of sex differences in the mechanisms underlying larval growth, and in the dopaminergic system, the authors should separate the sexes to determine whether males and females show different responses to loss of STIM in THD’ calcium responses (and other heterogeneous phenotypes).

2. The precise genotypes are not always listed in figures or in the text. Complete genotypes should always be available to readers for the final publication in all those locations.

3. The authors should cite the Britton paper from 2002 in which they show that changes in insulin pathway activity cause the animals to leave the food, as that is a relevant finding to the authors’ potential identification of ilp3 in regulating feeding.

4. Line 55 tachykinin is misspelled.

5. There are many feeding studies and hormones that affect feeding not cited in the introduction (including but not limited to ecdysone, Akh, etc). Please include a better more detailed introduction to the rich literature on this topic, as there are many studies.

6. Line 74 – I don’t agree with the statement that circuits that maintain feeding are not well understood. I think there is a lot of literature in this space, so the authors should moderate this claim.

7. Line 106 – I don’t agree with the statement that the small size of the STIMKO must be due to a deficit in cell division. Many larval tissues are endoreplicating, so they grow but don’t really divide. Also, there are many nonautonomous effects of specific cell types like the fat body on growth that don’t involve problems in cell division. The authors will need to adjust the text for accuracy.

8. Because the authors show that there are defects in neuroblast cell divisions, they will need to quantify the number of THD’ cells in STIMKO mutants. Currently they say the THD’ cells “appeared no different in STIMKO…” – this claim must be backed up by data given the neuroblasts defects.

9. The colors in the images should be changed from red and green to magenta and green to be more accessible to individuals with visual impairments.

10. Proving a physiological role for STIM in THD’ neurons should include evidence that the gene is normally expressed there.

11. Please state your food recipe, as this is essential information in replicating larval experiments.

12. Please reference Delanoue et al 2009 for the pupal volume calculation.

13. Please acknowledge Flybase and its funding, the Bloomington Drosophila stock center and its funding, and VDRC and its funding.

**Have all data underlying the figures and results presented in the manuscript been provided?**

Reviewer #1: Yes

Reviewer #2: Yes

Reviewer #3: **No: **There is RNAseq data that is mentioned as rationale for doing a series of experiments on ilp3 that is never presented or analyzed at all "will be published in future". For publication in an open-access journal, to me this is unacceptable.

PLOS authors have the option to publish the peer review history of their article (what does this mean?). If published, this will include your full peer review and any attached files.

Reviewer #1: No

Reviewer #2: No

Reviewer #3: No

---

## [Decision Letter · Decision Letter 1]

7 May 2023

Dear Gaiti,

Thank you very much for submitting your Research Article entitled 'A STIM dependent dopamine-insulin axis maintains the larval drive to feed and grow in Drosophila' to PLOS Genetics.

The revised manuscript was seen by reviewers #1 and #3 of the original submission, who have recommended major and minor revision, respectively. The manuscript, reviews, and review history have now been discussed among senior editors. Overall, we think that an additional round of minor revision will be needed in order to move forward. There are a number of specific points raised by both reviewers that we think will be helpful as you prepare your revision. In particular, we ask that you pay careful attention to concerns regarding over-interpretation and assumptions, ensuring consistency between the data presented and the conclusions of the manuscript, as outlined in comments from reviewer #1 (paragraph 2). 

We therefore ask you to modify the manuscript according to the review recommendations. Your revisions should address the specific points made by each reviewer.

We hope to receive your revised manuscript within the next 60 days. If you anticipate any delay in its return, we would ask you to let us know the expected resubmission date by email to plosgenetics@plos.org.

Yours sincerely,

Gregory S. Barsh

Editor-in-Chief

PLOS Genetics

Gregory Copenhaver

Editor-in-Chief

PLOS Genetics

Reviewer's Responses to Questions

**Comments to the Authors:**

**Please note here if the review is uploaded as an attachment**.

Reviewer #1: The findings on the STIM mutation, its site of action (a few TH neurons), and its systemic impact are very interesting and intriguing. The additional data on the neuropeptide Ilp3, TH and dopamine receptors are informative.

However, the central question remains unresolved whether THD's dopaminergic neurons regulate larval growth and feeding independently or whether the growth defect is a consequence of feeding, as the authors suggest, and how the two processes may be linked or mediated. This uncertainty persists because silencing the Ilp3 peptide, upregulated in STIM_KO larvae, partially rescues the growth defect and viability without affecting feeding, indicating that these processes may be independent. The finding that Ilp3 is upregulated in STIM-KO also raises questions about whether the growth defect is caused by a nutrient deficiency or the presence of abnormal levels of Ilp3 and/or other factors that arrest the developmental progression and further growth.

Some specific comments:

The authors declare in their rebuttal that Ilp3 does not affect feeding rate and "..... We have appropriately modified our conclusions to reflect these data." However, the revised paper does not consistently update its conclusions throughout the text, and the model is still based on assumptions requiring further work. For instance, the original model and conclusions are still reflected in the title, even though they are not supported by the current data. Likewise, in the authors' summary, the following text: "The identified dopaminergic neurons dysregulate the normal pattern of larval ilp3 expression leading to premature cessation of feeding and growth" it may suggest that Ilp3 mediates both effects, which creates false expectations for readers. In line 103, "We identified a novel dopaminergic-neuropeptide connection that drives persistent feeding and growth in early larvae." The study does not identify the neuropeptide(s) downstream of the dopaminergic neurons associated with persistent feeding, and the role in growth remains incompletely characterized.

Figure 1E shows the measurement of larval length at specific chronological ages. The figure is informative. However, it should be noted that these measurements only reflect the cumulative increase in size over time and should not be confused with growth rate, which is measured differently with a defined formula. Relevantly, given the significant delay of STIM_KO mutants entering L2 and L3 around 20-24 or more hours later than the control, the authors must compare developmental age, not chronological age. With such delay, the chronological age is irrelevant. Based on the authors' carefully collected data on chronologically aged larvae in Figure 1, the control larvae at 58-62 h are in L2, while the STIM_KO larvae only enter the L2 at 72-78 h. Hence, the 72-74 h STIM_KO larvae are of a developmental age approximately that of the control larvae at 58-62 h. Few differences are noticed by comparing the sizes of 58-62 h larvae (control) with the size of 72-74 h (mutant). The mutant larvae appear severely delayed in their development but arrive at approximately the expected size of the L1 and L2. Yet, as observed by the authors, the larvae do not seem to progress or grow further, and larvae size decreases after 84 h.

The mutant larvae already experience a feeding impairment from their early stages of development, which could contribute to their delayed growth. The mutant larvae require additional time, but they seem to reach the appropriate size for moulting. Once feeding ceases, the larvae become arrested and cannot continue growing, and the water loss may explain the decline in body weight or length.

In this case, comparing the length of larvae between animals of the same chronological age is inaccurate. By using this method, the authors may come to incorrect conclusions.

The role of Ilp3 is important for understanding STIM defects. The authors suggest that Ilp3 acts as an antigrowth signal based on animals' larger and smaller sizes when Ilp3 is manipulated. This idea counters the Ilp3-KO mutant defect. "...mutants lacking Dilp3 develop with normal timing and show

only a modest reduction in fecundity" (https://doi.org/10.1038/ncomms7846). An analysis using endogenous Ilp3

mutations must clarify this point. Moreover, an increase in size may result from an increase in growth rate or a longer growth time at a normal or reduced growth rate. Therefore, caution must be exercised without proper analysis since there are numerous examples of mutants with larger pupae and adults due to developmental extension and delay without such factors being an antigrowth signal. In the case of dsIlp3 animals, the significant delay in their development may explain the increase in the size of their larvae and pupae, either in part or in whole, through an indirect effect on size due to delayed maturation time. Overexpression of Ilp3 also leads to developmental arrest and delay. If Ilp3 inhibits growth, the growth rate should increase relative to developmentally age-matched controls, and developmental time should be accelerated rather than delayed —as seen in SDR mutations with clear increases in growth rate. However, if growth is not the primary effect, larval growth is not expected to increase above age-matched controls. To affirm the antigrowth action of Ilp3, the authors must measure larval growth rate over time and compare it properly.

Starved animals (which grow slowly or do not grow) decrease rather than increase Ilp3 transcription (Ikeya et al., 2002) and increase other ILPs such as ILP6. If the authors are correct, STIM-deficient larvae with

reduced feeding should show signs of starvation. The authors' data on Ilp genes presented in the article do not support with Ilp3 increased and ILP6 unchanged. The upregulation of Ilp3 transcription, despite being on slow feeding, is intriguing and does not add well to the model proposed by the authors. The model seems too simplistic.

This reviewer suggested L-DOPA supplementation experiments clarify the role of dopamine downstream of THD' STIM-KO. Still, the authors justify not doing the experiment by arguing that the mutants are defective in feeding and would not work. Surprisingly, their Discussion describes that such an experiment works in mice with a feeding defect due to a lack of dopamine. Lines 506-509 "where prenatal mice genetically deficient for dopamine (DA-/-) could not feed and died from starvation. Feeding could, however, be initiated upon supplementation with L-DOPA (64), allowing them to survive. "

The experiments with TH-RNAi and dopamine receptors in limited neurons are informative but insufficient. The TH-RNAi defect recapitulates some aspects of the STIM defect. Still, it does not mimic the effects of STIM deficiency, which is lethal, so there is more than suppressing the dopaminergic pathway. L-DOPA supplementation could clarify the impact of impaired feeding and what are causes and consequences. The observation that none of the manipulations causes the lethality observed in STIM deficiency cannot be solely explained by RNAi not being a complete knockout because the lethality is seen in STIM-RNAi. The STIM defect is more complex than the model suggests.

Furthermore, the authors propose, "In summary, we have identified a novel STIM-dependent dopamine-ILP circuit that regulates developmental changes in larval feeding behaviour"; however, such as instructive role is an overinterpretation. An instructive role in coordinating developmental feeding behaviour (e.g. rate of feeding and stop feeding) must be demonstrated by showing that STIM can drive larvae to eat when they stop eating (at wandering) or feed faster and grow faster. The authors are also mistaken in describing that wandering larvae feeding stops gradually. Feeding slows gradually after 96 h and stops entirely as larvae wander out of the food to pupate.

Minor issues:

• Figure 4 shows that the dendrites are labelled with anti-RFP. This antibody is not included in the Methods. Although mCherry is a red fluorescent protein, describing the DenMark (mCherry) construct in the Resource Table is more appropriate (see flyBase and ref .40)

• Syt-eGFP.

• MNSc is described in different ways, including "MNC"; the original description was mNSC.

• "Quantification of the weight of 10 flies from the indicated genotypes". The sex of the flies must be indicated.

• The authors state, "synthesis and release of the steroid hormone 20-OH ecdysone from the prothoracic gland … (16,17). This statement is incorrect. The prothoracic gland releases pro-ecdysone (alpha-ecdysone), converted peripherally to 20-OH ecdysone and other ecdysteroids.

• The Discussion pertains to juvenile animals, but the references to the insulin-dopaminergic axis in other species relate to adult animals. This must be explicitly stated in the text or compared to a parallel system involving juvenile animals or children. References 68 and 69 predominantly concern the role of insulin in dopamine modulation, so the. A suggestion of conservation between the two is misleading.

In summary, authors should review their manuscript and ensure consistency between their conclusions and presented data, interpreting the data correctly and without overinterpretation or assumptions regarding mutant behaviour without supporting data. This is important because the previous version had some unsupported assumptions, and therefore the description of the phenotype should be moderated throughout the manuscript according to the available data. The phenotype and data remain intriguing, but it is unclear whether it is simply a feeding defect that becomes chronic and causes other alterations, leading to growth arrest and death, or if the mutation additionally affects systemic processes that influence growth along with persistent feeding. The role of the THD neurons and dopamine in normal development and changes in growth patterns throughout development is unclear.

Reviewer #3: I thank the authors for their efforts to address my concerns. While the authors have addressed several concerns in part, they have not fully addressed them.

Major comments

1. Following up on my point in the original review about using the correct genotypes in doing a rescue experiment. In their response to reviewers the authors state they used the genotypes I set out in my original review in the bar graph in Figure S2A, and have “performed the appropriate genetic controls for all other genotypes tested and have included them either in the main figure or the associated supplementary figure.”

Perhaps this information was missing somehow from the figures but I can only find the appropriate genotypes that must be used when doing all rescue experiments in Figure S2A. As far as I can tell from the material to which I have access, the rescue experiments simply use Canton S, the StimKO, and then the rescue genotype. These are not the correct control genotypes for the reasons I set out in my original review.

At this point, the authors will need to moderate their conclusions in these other experiments by saying “as far as we can tell from the control genotypes we used, there was a rescue; however, future experiments will need to verify this conclusion by showing the UAS and GAL4 lines alone do not provide any significant rescue in this phenotype”.

2. I am still unsure why the RNAseq data analysis was not shown and why this technique and analysis workflow was not included in the methods. For readers and reviewers to evaluate this data the workflow must be presented and the data shown including a statistical analysis.

3. In Line 449 the authors draw a direct line between the THD’ neurons, the feeding deficit, and the MNSCs – their feeding data with dilp3 contradicts this sentence and it will need to be modified.

**Have all data underlying the figures and results presented in the manuscript been provided?**

Reviewer #1: Yes

Reviewer #3: Yes

PLOS authors have the option to publish the peer review history of their article (what does this mean?). If published, this will include your full peer review and any attached files.

Reviewer #1: No

Reviewer #3: No

---

## [Editor Report · Decision Letter 2]

11 Jun 2023

Dear Dr Hasan,

We are pleased to inform you that your manuscript entitled "A STIM dependent dopamine-neuropeptide axis maintains the larval drive to feed and grow in Drosophila" has been editorially accepted for publication in PLOS Genetics. Congratulations!

The senior editors believe that the minor changes made in response to the previous round of reviews satisfactorily address the reviewers' (and editors') concerns, and we look forward to publishing the work.

Yours sincerely,

Gregory S. Barsh

Editor-in-Chief

PLOS Genetics

Gregory Copenhaver

Editor-in-Chief

PLOS Genetics

Comments from the reviewers (if applicable):

**Data Deposition**

http://datadryad.org/submit?journalID=pgenetics&manu=PGENETICS-D-22-01075R2

**Press Queries**

---

## [Editor Report · Acceptance letter]

22 Jun 2023

PGENETICS-D-22-01075R2 

A STIM dependent dopamine-neuropeptide axis maintains the larval drive to feed and grow in Drosophila 

Dear Dr Hasan, 

We are pleased to inform you that your manuscript entitled "A STIM dependent dopamine-neuropeptide axis maintains the larval drive to feed and grow in Drosophila" has been formally accepted for publication in PLOS Genetics! Your manuscript is now with our production department and you will be notified of the publication date in due course.

With kind regards,

Zsofia Freund

PLOS Genetics

On behalf of:
